# POC: Preventing the Over-Collapse of Classes for Class-Incremental Learning

## Abstract

Deep neural network-based classification models often suffer from catastrophic forgetting during class-incremental learning (CIL). Previous studies reveal that it results from the overlap between seen and future classes after being mapped by model to its feature space through extracting the features. In this paper, we analyze that this overlap mainly results from the *over-collapse* of seen classes, where the model tends to map originally separated one seen class and its adjacent regions in input space to be mixed in the feature space, making them indistinguishable. To this end, we propose a two-step framework to **P**revent the **O**ver-**C**ollapse (POC). During training, POC first learns and applies a set of transformations to the training samples of seen classes. Based on our theoretical analysis, the transformation results will locate in the adjacent regions of the seen classes in the input space so that we can let them represent the adjacent regions. Then, the model's optimization objective is modified to additionally classify between the seen classes and the adjacent regions, separating them in model's feature space so that preventing the over-collapse. To retain the model's generalization on the seen classes, a deterministic contrastive loss that makes the separate features of seen classes and adjacent regions close is further introduced. Since POC uses the adjacent regions exclusively for classification, it can be easily adopted by existing CIL methods. Experiments on CIFAR-100 and ImageNet demonstrate that POC effectively increases the last/average incremental accuracy of six SOTA CIL methods by 3.5%/3.0% on average respectively.

## 1 Introduction

Over the past few years, incremental learning (IL) has attracted extensive attention to facilitate a model learning from a sequence of tasks. Within class-incremental learning (CIL), each task centers on image classification and introduces new classes. The primary goal of CIL is to develop a unified classification model that maps an input into a feature space by extracting its features and then classifies it among all of the encountered classes with the features. However, if the model is solely fine-tuned for each new task, it will suffer from a severe problem known as catastrophic forgetting, where its knowledge of the old tasks fades and the performance degrades greatly (McCloskey & Cohen, 1989).

Various approaches have emerged to address catastrophic forgetting, broadly categorized into three groups: replay-based (Bang et al., 2021; Iscen et al., 2020; Lin et al., 2023; Rolnick et al., 2019; Tiwari et al., 2022), regularization-based (Aljundi et al., 2018; 2019; Jung et al., 2020; Sun et al., 2023), and architecture-based (Aljundi et al., 2017; Douillard et al., 2022; Li et al., 2019; Pham et al., 2021; Yoon et al., 2018) methods. Despite their efficacy in CIL, the primary optimization objective of existing methods, typically a classification loss among the seen classes, has overlooked the necessity of learning a representation compatible with future classes. Specifically, as depicted in Figure 1(a), their optimization objective will lead to the *over-collapse* of seen classes, where the model will map originally separated one seen class and its adjacent regions in the input space to be mixed in the feature space in order to improve model's generalization on the seen class. Although this phenomenon is observed, analyzed and desired in other classification tasks (Li et al., 2018; Fawzi et al., 2018), its effect is not considered within the CIL scenario yet. However, as shown in Figure 1(a) and our experiments, the over-collapse will increase the risk of overlap between the areas covered by the seen and future classes after being mapped into model's feature space, making them indistinguishable. As stated by Masana et al. (2022), this overlap can result in knowledge forgetting of seen classes since the model will misidentify the samples of seen classes belonging to future classes after learning.

In this paper, we propose a two-step framework to **P**revent the **O**ver-**C**ollapse (POC) to bolster CIL performance. During training, POC learns and applies a set of transformations to the training samples of seen classes. Based on our theoretical analysis and findings of prior works (Tack et al., 2020), the transformation results will locate in adjacent regions of seen classes in input space so that we can let them represent adjacent regions. Then, the model is expanded with another classifier and its optimization objective is modified to additionally classify between seen classes and adjacent regions, separating them in model's feature space. In this way, the over-collapse is prevented so that future classes are protected from overlap, avoiding catastrophic forgetting. During testing, the expanded classifier is masked, enabling the model to classify between seen classes without introducing extra computation burden. Since the adjacent regions are exclusively leveraged for classification, POC can be easily adopted by existing CIL methods to improve their performance. Moreover, because the adjacent regions are semantically similar to their original classes, classifying between them without additional constraints can make them distant in model's feature space (Elsayed et al., 2018) and harm the model's generalization. Therefore, a deterministic contrastive loss is introduced to make the adjacent regions close to their original classes in the feature space. In this way, the generalization of the model on the seen classes is protected, improving performance.

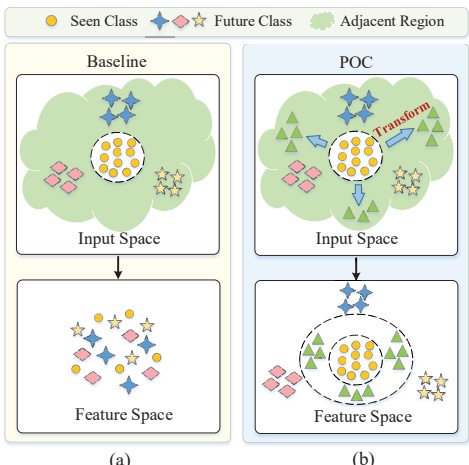

Figure 1: Illustration of POC. In (a), a baseline model will map originally separated one seen class and its adjacent regions in the input space to be mixed in the feature space, leading to the over-collapse of the seen class. The areas covered by the seen and future classes in model's feature space will then overlap, causing catastrophic forgetting. Contrastingly, in (b), POC produces samples in the adjacent regions by transformations and trains the model to classify between adjacent regions and seen class, preventing the over-collapse.

In summary, our contributions are three-fold:

- We analyze the over-collapse phenomenon and its negative effect for class-incremental learning (CIL). Therefore, we propose to **P**revent the **O**ver-**C**ollapse (POC) to protect future classes from overlapping with seen classes in model's feature space for CIL.
- POC can be easily adopted by existing CIL methods to improve their performance. Moreover, a deterministic contrastive loss protecting the generalization of the model on the seen classes is introduced for POC, further improving its effectiveness.
- Experiments on CIFAR-100 and ImageNet show that POC increases the last/average incremental accuracy of six SOTA CIL methods by 3.5%/3.0% on average respectively.

## 2 RELATED WORK

### 2.1 CLASS-INCREMENTAL LEARNING

Class-incremental learning (CIL) strives to enable a unified model to learn sequentially from different image classification tasks while retaining knowledge from previous ones (Hung et al., 2019; Singh et al., 2020; Wang et al., 2022; Yan et al., 2021; Zhou et al., 2021). Generally, the methods fall into three categories: (1) replay-based; (2) regularization-based; (3) architecture-based.

Replay-based methods (Bang et al., 2021; Hou et al., 2018; Wu et al., 2019) offer an intuitive solution by storing samples from previous tasks in a memory buffer. When learning new tasks, the stored samples are amalgamated with new data to train the model. Initially, Riemer et al. (2018) used an equal proportion of new and stored samples to calculate the loss. Douillard et al. (2020) and Hou et al. (2018) further utilized stored samples to calculate a distillation loss for knowledge transfer.

To mitigate memory usage for storing samples, regularization-based methods (Lopez-Paz & Ranzato, 2017; Wang et al., 2021) augment the classification loss with a regularization term, preventing crucial parameters from changing too much. For instance, Aljundi et al. (2018) quantified parameter

importance for old tasks, then regularized alterations of significant parameters. Alternatively, Li & Hoiem (2017) employed knowledge distillation to maintain logits of new samples on old classes.

Architecture-based methods offer an alternative approach by modifying the model or maximizing feature distance across different tasks, making different tasks mutually independent. In the work of Rajasegaran et al. (2019) and Serra et al. (2018), upon the arrival of a new task, a specifc part of the model is trained and then fixed. In the work of Chaudhry et al. (2020), Tang et al. (2021) and Xiang & Shlizerman (2023), the authors designed special losses to maximize the distance across tasks.

A shared problem of the aforementioned methods is their exclusive emphasis on enhancing discrimination among the seen classes during current task learning. However, this singular focus will lead to the over-collapse of seen classes, causing catastrophic forgetting. Instead, with seen classes, our POC produces samples in their adjacent regions and trains the model to classify between them and the seen classes. This strategy can prevent the over-collapse, making the model adaptable to future tasks.

While similar to FACT (Zhou et al., 2022) and IL2A (Zhu et al., 2021) in generating extra data for training, our POC differs in key aspects. Their foundational principle is to train the model to classify between seen and future classes in advance. It is based on assumption that positions of future classes can be predicted with the seen classes using mixup. However, future classes vary a lot and are unpredictable. Differently, POC aims to enhance the feature's discrimination by producing samples in adjacent regions of seen classes and training the model to classify between these regions and seen classes. In this way, the over-collapse is prevented and the future classes will be naturally protected from overlapping, regardless of their actual positions. Detailed experiments are in Section 4.2.2.

### 2.2 OUT-OF-DISTRIBUTION DETECTION

Out-of-distribution (OOD) detection (Bai et al., 2021; Cai & Fan, 2022; Jeong & Kim, 2020) aims at predicting whether a sample aligns with the training data. In a specific approach (Cai & Fan, 2022; Golan & El-Yaniv, 2018; Mohseni et al., 2021; Tack et al., 2020), researchers pre-trained a model to predict the geometric transformation on input samples. It was shown that the transformed samples are OOD-like, while similar to their original samples (Tack et al., 2020). It inspires us to learn a set of transformations whose application results are adjacent to the seen classes. By classifying them from the seen classes, their features will be separated so that the over-collapse will be prevented.

## 3 METHODOLOGY

This section provides a comprehensive overview of our proposed POC. Section 3.1 briefly formulates the CIL problem alongside existing methods, followed by the motivation of POC. Section 3.2 details the framework of POC, emphasizing its approach to learn a set of transformations to produce samples locating in so that representative of adjacent regions of seen classes in input space and modify model's optimization objective to additionally classify between seen classes and adjacent regions, separating them after being mapped by model into its feature space. This strategy prevents the over-collapse and protects future classes from overlapping with seen classes in model's feature space.

### 3.1 PROBLEM STATEMENT AND MOTIVATION

In CIL, a model will learn from $\mathcal{T}$ tasks and should classify an input among all seen classes at any time step. Specifically, when learning from task $t$, $L_t$ new classes $C^t = \{c_i^t\}_{i=1}^{L_t}$ are introduced and the model has only access to the new training dataset, while the evaluation is performed on the union of the testing datasets encountered so far. In general, the model consists of two primary components: a feature extractor $f : \mathcal{X} \to \mathbb{R}^d$ that maps an input into a feature space by extracting its features and a classifier $\Phi : \mathbb{R}^d \to \mathbb{R}^{\widetilde{L}_t}$ that calculates the probability distribution over seen classes with the extracted features, where $\widetilde{L}_t = \sum_{i=1}^t L_i$. CIL aims to obtain better performance after learning $\mathcal{T}$ tasks.

Most of the existing CIL methods can be formulated into the following paradigm. Given the training samples $\{\{(x_{i,j}, y_i)\}_{j=1}^{N_i}\}_{i=1}^{\widetilde{L}_t}$, where $y_i \in \cup_{k=1}^t \{C^k\}$ since there can be a memory buffer storing a subset of samples from the old tasks, a classification loss $\mathcal{L}_{\text{Cls}}$ is calculated:

$$\mathcal{L}_{\text{Cls}} = \sum_{i=1}^{\widetilde{L}_t} \sum_{j=1}^{N_i} \mathcal{L}(\Phi(f(x_{i,j})), y_i), \tag{1}$$

Figure 2: Overview of our POC. During training, it operates in two steps "Adjacent Region Labeling" and "Collapse Prevention". First, a set of transformations are learned and applied to training samples of seen classes to produce samples locating in so that representative of their adjacent regions in the input space. Then, the model is expanded with another classifier and its classification loss is modified to additionally classify between the seen classes and adjacent regions, preventing the over-collapse. A deterministic contrastive loss is introduced to preserve model's generalization. During testing, the expanded classifier is masked to leverage the original model to classify among the seen classes.

where $\mathcal{L}$ represents a classification criterion, e.g. cross-entropy loss. Additionally, a regularization loss $\mathcal{L}_{\text{Reg}}$ may be employed to preserve important parameters or transfer knowledge. These two losses are combined to optimize the model. However, as stated in Section 1 and shown in experiments, this training paradigm will lead to over-collapse so that seen and future classes will overlap after being mapped into model's feature space, which will result in catastrophic forgetting (Masana et al., 2022).

Drawing from our observation, we propose to first produce samples locating in so that representative of $n$ adjacent regions of the seen classes in input space $\{\{\{(\hat{x}_{i,k,m}, l_{i,k})\}_{m=1}^{M_{i,k}}\}_{k=1}^{n}\}_{i=1}^{\widetilde{L}_t}$. Subsequently, we modify the classification loss to be $\mathcal{L}_{\text{Mod\_Cls}}$ to incorporate classification between these adjacent regions and the seen classes, separating them in model's feature space to prevent the over-collapse:

$$\mathcal{L}_{\text{Mod\_Cls}} = \sum_{i=1}^{\widetilde{L}_t} \sum_{j=1}^{N_i} \mathcal{L}(\Phi(f(x_{i,j})), y_i) + \sum_{i=1}^{\widetilde{L}_t} \sum_{k=1}^{n} \sum_{m}^{M_{i,k}} \mathcal{L}(\Phi(f(\hat{x}_{i,k,m})), l_{i,k}), \tag{2}$$

### 3.2 PREVENT THE OVER-COLLAPSE

With the above purpose, we propose POC that can be easily adopted by the CIL methods to help address the problem and improve their performance. As illustrated in Figure 2, POC works in two steps "Adjacent Region Labeling" and "Collapse Prevention" during training. In the first step, POC learns and applies a set of transformations to the training samples of seen classes to produce samples in and representative of their adjacent regions in the input space. In the second step, the model is expanded with another classifier and its optimization objective is modified to additionally classify between the adjacent regions and the seen classes, separating them in model's feature space so that preventing the over-collapse. Furthermore, to protect the generalization of the model on the seen classes, we introduce a deterministic contrastive loss that makes the separate features of the seen classes and adjacent regions close. During testing, the expanded classifier is masked, allowing the original model to classify inputs among seen classes without introducing additional inference costs.

### 3.2.1 ADJACENT REGION LABELING

To prevent the over-collapse, an intuitive first step is to generate samples in the adjacent regions of the seen classes in the input space and let them represent the adjacent regions. Initially, we experimented with applying mixup, rotation and other transformations to the training samples of seen classes to generate desired samples. However, we found that the resulting samples were too distant from or close to the original ones, unable to represent the adjacent regions and leading to suboptimal performance. Inspired by out-of-distribution (OOD) detection works (Cai & Fan, 2022; Golan & El-Yaniv, 2018; Mohseni et al., 2021; Tack et al., 2020), we propose to learn a new set of transformations to generate suitable samples. To substantiate this approach, we begin by proving that affine transformations have the capability to generate OOD samples through the following proposition:

**Proposition 3.1.** *Denote the distribution of images {x} as $P_r(x)$. For $\forall 0 < \beta < \max(P_r(x)), 0 < \delta < 1, \exists A_a \in \{A | A$ is the matrix of one affine transformation}, s.t. $P(P_r(A_a x) \leq \beta) \geq \delta$.*

*Proof.* Given in Appendix A.1. □

Second, we demonstrate that applying rotations to images results in transformed outputs whose distance from the original distribution is bounded by an upper limit.

**Proposition 3.2.** *Denote the probability distribution of images {x} from a specific class as $P(x)$. For $\forall A \in \{A_r | A_r$ is the matrix of one rotation transformation}, $\mathcal{W}(P(Ax), P(x)) \leq \delta$, where $\mathcal{W}$ is the Wasserstain distance and $\delta$ signifies a constant upper bound.*

*Proof.* Given in Appendix A.2. □

However, when directly using rotations, since some rotated samples, such as a rotated pen, have the same semantic as their original samples, the distribution of the rotated samples can overlap with that of the original samples. In this way, less adjacent regions are covered, reducing the effectiveness of POC as shown in Section 4.2.4 and C.1. Moreover, classifying the overlapped region into different classes is improper and is harmful for model training. To take advantages of rotations according to **Proposition** 3.2 and according to **Proposition** 3.1, we propose to learn a set of affine transformations whose parameter matrices are denoted as follows:

$$\{\theta_i\}_{i=1}^n = \left\{ \begin{pmatrix} p_{i,1} & p_{i,2} & p_{i,3} \\ p_{i,4} & p_{i,5} & p_{i,6} \end{pmatrix} \right\}_{i=1}^n.$$

We then calculate a transform loss $\mathcal{L}_{\text{Trans}}$ to make the transformations similar to but not the same as the rotations so that the transformed results will be OOD while close to original samples:

$$\mathcal{L}_{\text{Trans}} = \sum_{i=1}^n (\sum_{j=1}^2 \sum_{k=1}^3 (\theta_i - \widehat{\theta}_i)_{j,k}),$$

$$s.t. \ \{\widehat{\theta}_i\}_{i=1}^n = \left\{ \begin{pmatrix} \cos \frac{2\pi i}{n} & -\sin \frac{2\pi i}{n} & 0 \\ \sin \frac{2\pi i}{n} & \cos \frac{2\pi i}{n} & 0 \end{pmatrix} \right\}_{i=1}^n. \tag{3}$$

Complemented by the modified classification loss detailed in Section 3.2.2, $\mathcal{L}_{\text{Trans}}$ will optimize the parameters of transformations to make the transformed results locate in and representative of adjacent regions of seen classes in the input space, quantitatively shown in Section 4.2.4. Our experiments in Section C.3 underscore that the acquired transformations significantly enhance POC's performance compared to mixup and other transformations. Based on these parameters, the learned transformations are then applied to the training samples of the seen classes, generating a set of new samples $\{\{\{(x_{i,j,k}, y_{i,k})\}_{j=1}^{N_i}\}_{i=1}^{\widetilde{L}_t}\}_{k=1}^n$, where $x_{i,j,k}$ is produced by applying $k$-th transformation to $x_{i,j}$ and $y_{i,k}$ is a generated new class. In this way, $\{\{(x_{i,j,k}, y_{i,k})\}_{j=1}^{N_i}\}_{k=1}^n$ are considered as $n$ labeled adjacent regions of class $y_i$ in input space produced with its training samples $\{(x_{i,j}, y_i)\}_{j=1}^{N_i}$.

### 3.2.2 COLLAPSE PREVENTION

With the labeled adjacent regions, to prevent the over-collapse, a direct solution is then training the model to classify between adjacent regions and seen classes. Therefore, besides $\Phi$, we introduce another classifier $\overline{\Phi}: \mathbb{R}^d \rightarrow \mathbb{R}^{n\widetilde{L}_t}$, which also takes the extracted features of $f$ as input and calculates the probability over the adjacent regions. With the transformed training samples labeling adjacent regions and additional classifier, the classification loss $\mathcal{L}_{\text{Cls}}$ will be modified to be $\mathcal{L}_{\text{Mod\_Cls}}$ as follows:

$$\mathcal{L}_{\text{Mod\_Cls}} = \frac{n}{n+1} \sum_{i,j} \mathcal{L}(\Phi \circ \overline{\Phi}(f(x_{i,j})), y_i) + \frac{1}{n+1} \sum_{i,j,k \neq 0} \mathcal{L}(\Phi \circ \overline{\Phi}(f(x_{i,j,k})), y_{i,k}), \tag{4}$$

where $\circ$ means concatenating the outputs of two classifiers. $\mathcal{L}_{\text{Mod\_Cls}}$ will substitute $\mathcal{L}_{\text{Cls}}$ to train the model to make the seen classes and their adjacent regions separated in the feature space, preventing the over-collapse. During testing, the classifier $\overline{\Phi}$ is masked so that only $\Phi$ is used to calculate the probability distribution over the original seen classes, without introducing additional cost.

### 3.2.3 Deterministic Contrastive Loss

The proposed two steps separate the features of seen classes and adjacent regions without constraining their distance, which will make the model map the seen classes and adjacent regions to be distant from each other in the feature space for better classification. However, since the training samples cannot completely represent their classes, an actual testing sample can locate in the adjacent regions produced with training samples. In this way, the model's generalization on the seen classes will be impeded because the model will map these testing samples to be far from the training samples in the feature space, crossing the classification boundary and leading to wrong classification result. To this end, we introduce a deterministic contrastive loss that makes the features of the seen classes and adjacent regions close. In addition, to maintain the diversity of transformations to cover more adjacent regions, the features of adjacent regions are kept away from each other. Specifically, for each training sample $x$, after applying the transformations to it, we can get a set of results $\{x_i\}_{i=0}^n$, where $x_i$ is produced by $i$-th transformation and $x_0$ equals $x$. With the feature extractor $f$, we can further get the features of these results $\{f(x_i)\}_{i=0}^n$ and define the similarity between two results:

$$\text{sim}(x_i, x_j) = \exp(\psi(f(x_i), f(x_j))/\tau), \tag{5}$$

where $\psi(a, b) = \frac{a \cdot b}{||a||||b||}$ calculates the cosine similarity between the inputs and $\tau$ is a temperature parameter adjusting the scale. With the defined similarity, the deterministic contrastive loss for one training sample is calculated as follows and then averaged over all samples:

$$\mathcal{L}_{\text{DCL}} = -\sum_{i=1}^n \log \frac{\text{sim}(x_i, x_0)}{\text{sim}(x_i, x_0) + \sum_{j \neq i} \text{sim}(x_i, x_j)}. \tag{6}$$

### 3.2.4 Total Objective

During training, we aggregate the aforementioned optimization objectives to form the total loss $\mathcal{L}_{\text{Total}}$:

$$\mathcal{L}_{\text{Total}} = \mathcal{L}_{\text{Mod\_Cls}} + \mathcal{L}_{\text{Reg}} + \lambda_1 \mathcal{L}_{\text{Trans}} + \lambda_2 \mathcal{L}_{\text{DCL}}, \tag{7}$$

where $\lambda_1$ and $\lambda_2$ are hyperparameters used to balance the scale of different losses. The model is then optimized with $\mathcal{L}_{\text{Total}}$, preventing the over-collapse and protecting the future classes from overlapping with the seen classes in model's feature space to make the model compatible with future tasks.

## 4 Experiments

### 4.1 Experimental Setup

#### 4.1.1 Datasets and Evaluation Metrics

In line with prior works, we choose CIFAR-100 (Krizhevsky et al., 2009), ImageNet-100 and ImageNet (Deng et al., 2009) for evaluation in experiments. ImageNet-100 is a subset of ImageNet with 100 random classes chosen according to the same principle of LUCIR (Hou et al., 2019). We use two metrics to evaluate the performance of one method, which are the last accuracy and the average incremental accuracy. Denoting the model's classification accuracy after learning from task $t$ as $\mathcal{A}_t$, then the last accuracy is defined as $\mathcal{A}_{\mathcal{T}}$. For average incremental accuracy, it is calculated as $\frac{1}{\mathcal{T}} \sum_{t=1}^{\mathcal{T}} \mathcal{A}_t$, indicating the performance of the model along the whole learning procedure. The data unit of the reported results is "%".

To further quantify the distance between the seen and future classes and the generalization of the model on the seen classes, we define another two metrics named inter-class distance (ICD) and intra-class generalization (ICG). After task $t$ and before learning from task $t+1$, we collect the training samples of the seen and newly introduced classes and the testing samples of the seen classes. Then we use the feature extractor $f$ to obtain the features of each sample and calculate the mean of the features for each class. Assuming that the means of the features for the seen classes and new classes are $\{\mu_{i,\text{train}}\}_{i=1}^{\widetilde{L}_t}$, $\{\mu_{i,\text{test}}\}_{i=1}^{\widetilde{L}_t}$ and $\{\eta_i\}_{i=1}^{L_{t+1}}$ respectively, then the ICD and ICG are defined as follows:

$$\text{ICD} = \frac{1}{L_{t+1}} \sum_{i=1}^{L_{t+1}} \max(\{\psi(\eta_i, \mu_{j,\text{train}})\}_{j=1}^{\widetilde{L}_t}), \tag{8}$$

Table 1: Performance analysis of POC on CIFAR-100 under 6 task settings. "B" and "C" represent the class number of the first task and the following tasks respectively. The experiments are run for 3 times and the mean and variance of average incremental accuracy are reported.

| Method | Class Number Settings | | | | | |
| --- | --- | --- | --- | --- | --- | --- |
| | B = 50 | | | B = 20 | | |
| | C = 10 | C = 5 | C = 1 | C = 10 | C = 5 | C = 1 |
| LUCIR | 64.1±0.9 | 61.2±0.7 | 55.9±0.3 | 59.4±0.5 | 57.6±0.3 | 48.5±0.2 |
| w/ POC | **66.8(+2.7)±0.7** | **63.5(+2.3)±0.6** | **59.6(+3.7)±0.4** | **63.8(+4.4)±0.3** | **59.2(+1.6)±0.3** | **53.1(+4.6)±0.2** |
| CwD | 67.2±0.2 | 62.8±0.1 | 59.7±0.2 | 64.3±0.4 | 61.2±0.5 | 53.6±0.3 |
| w/ POC | **69.6(+2.4)±0.4** | **65.4(+2.6)±0.3** | **62.3(+2.6)±0.5** | **68.3(+4.0)±0.2** | **66.1(+4.9)±0.4** | **59.1(+5.5)±0.2** |
| PODNet | 64.6±0.7 | 63.2±1.1 | 59.8±0.5 | 54.9±0.4 | 53.2±0.4 | 50.5±0.2 |
| w/ POC | **68.2(+3.6)±0.8** | **67.2(+4.0)±1.0** | **63.1(+3.3)±0.7** | **60.6(+5.7)±0.7** | **58.3(+5.1)±0.4** | **53.5(+3.0)±0.5** |
| MEMO | 70.2±0.5 | 69.0±0.7 | 61.4±0.3 | 69.5±0.5 | 67.3±0.8 | 63.2±0.4 |
| w/ POC | **71.8(+1.6)±0.6** | **70.4(+1.4)±0.4** | **63.5(+2.1)±0.5** | **70.9(+1.4)±0.6** | **69.3(+2.0)±0.4** | **64.8(+1.6)±0.6** |
| LODE | 68.7±0.6 | 64.6±0.8 | 58.5±0.4 | 66.2±0.5 | 64.4±0.3 | 59.2±0.5 |
| w/ POC | **70.0(+1.3)±0.5** | **66.1(+1.5)±0.7** | **60.5(+2.0)±0.7** | **68.4(+2.2)±0.3** | **65.8(+1.4)±0.6** | **62.4(+3.2)±0.4** |
| MRFA | 68.0±0.4 | 66.4±0.6 | 60.3±0.8 | 67.8±0.8 | 65.7±0.6 | 61.3±0.7 |
| w/ POC | **69.2(+1.2)±0.4** | **68.1(+1.7)±0.5** | **62.7(+2.4)±0.5** | **69.6(+1.8)±0.6** | **67.5(+1.8)±0.4** | **63.6(+2.3)±0.8** |

Table 2: Performance analysis of POC on ImageNet-100 under 6 task settings. "B" and "C" represent the class number of the first task and the following tasks respectively and the last accuracy/average incremental accuracy are reported.

| Method | Class Number Settings | | | | | |
| --- | --- | --- | --- | --- | --- | --- |
| | B = 50 | | | B = 20 | | |
| | C = 10 | C = 5 | C = 1 | C = 10 | C = 5 | C = 1 |
| LUCIR | 61.4/71.5 | 55.1/67.2 | 41.1/56.8 | 48.0/61.5 | 42.6/55.7 | 34.3/48.9 |
| w/ POC | **64.0/73.7(+2.6/2.2)** | **57.6/68.3(+2.5/1.1)** | **47.7/61.8(+6.6/5.0)** | **51.5/65.2(+3.5/3.7)** | **46.4/59.3(+3.8/3.6)** | **36.9/51.5(+2.6/2.6)** |
| CwD | 60.4/71.6 | 55.8/68.2 | 40.3/56.3 | 48.2/62.9 | 44.6/58.5 | 34.3/51.1 |
| w/ POC | **62.3/73.2(+1.9/1.6)** | **57.4/69.4(+1.6/1.2)** | **44.7/59.8(+4.4/3.5)** | **51.2/64.4(+3.0/1.5)** | **47.1/60.6(+2.5/2.1)** | **38.9/53.1(+4.6/2.0)** |
| PODNet | 62.3/73.4 | 57.4/71.6 | 42.9/59.7 | 45.8/63.0 | 41.7/59.8 | 32.4/50.0 |
| w/ POC | **63.8/75.0(+1.5/1.6)** | **62.3/72.8(+4.9/1.2)** | **48.6/63.7(+5.7/4.0)** | **49.1/64.8(+3.3/1.8)** | **48.2/62.1(+6.5/2.3)** | **36.6/55.1(+4.2/5.1)** |
| MEMO | 66.2/76.8 | 64.5/76.4 | 52.7/64.0 | 53.6/67.1 | 48.4/60.8 | 40.3/53.2 |
| w/ POC | **67.4/77.9(+1.2/1.1)** | **66.5/77.8(+2.0/1.4)** | **55.9/66.5(+3.2/2.5)** | **55.4/68.2(+1.8/1.1)** | **50.7/62.5(+2.3/1.7)** | **42.4/54.7(+2.1/1.5)** |
| LODE | 64.5/73.6 | 59.4/71.0 | 45.8/60.4 | 50.6/63.5 | 45.3/59.5 | 37.2/52.1 |
| w/ POC | **66.1/75.1(+1.6/1.5)** | **61.7/73.1(+2.3/2.1)** | **50.4/63.8(+4.6/3.4)** | **53.4/65.7(+2.8/2.2)** | **48.3/62.3(+3.0/2.8)** | **40.5/53.4(+3.3/1.3)** |
| MRFA | 65.1/74.8 | 61.4/73.2 | 47.3/61.6 | 51.8/64.9 | 46.1/60.0 | 38.5/52.6 |
| w/ POC | **66.4/76.0(+1.3/1.2)** | **63.3/74.9(+1.9/1.7)** | **50.6/64.4(+3.3/2.8)** | **54.1/66.6(+2.3/1.7)** | **48.7/62.1(+2.6/2.1)** | **41.1/54.3(+2.6/1.7)** |

$$\text{ICG} = \frac{1}{\widetilde{L}_t} \sum_{i=1}^{\widetilde{L}_t} \psi(\mu_{i,\text{train}}, \mu_{i,\text{test}}), \qquad (9)$$

where $\psi(a,b) = \frac{a \cdot b}{||a||\,||b||}$ calculates the cosine similarity between the inputs. ICD measures the distance between each newly introduced class and its closest seen class, signifying the extent of overlap between them within model's feature space. A higher ICD value indicates increased overlap between new and seen classes. ICG assesses the distance between training and testing samples of each seen class, reflecting the model's generalization capacity on the seen classes. A higher ICG value signifies improved model generalization.

### 4.1.2 MODELS AND TRAINING

We incorporate our POC into six state-of-the-art CIL methods, which are LUCIR (Hou et al., 2019), CwD (based on LUCIR) (Shi et al., 2022), PODNet (Douillard et al., 2020), MEMO (Zhou et al., 2023), LODE (Liang & Li, 2023) and MRFA (based on FOSTER (Wang et al., 2022)) (Zheng et al., 2024). The classification models employed on CIFAR-100, ImageNet-100 and ImageNet are ResNet-32, ResNet-18 and ResNet-18. The hyperparameters $n$, $\tau$, $\lambda_1$ and $\lambda_2$ are set to be 3, 2, 10, 0.1 respectively. On CIFAR-100 and ImageNet-100, the methods will be evaluated across 6 task settings, whose class numbers of the first task and the following tasks are: (1) 50, 10; (2) 50, 5; (3) 50, 1; (4) 20, 10; (5) 20, 5; (6) 20, 1. As for ImageNet, 3 task settings are explored: (1) 500, 100; (2) 100, 100;

Table 3: Performance analysis of POC on ImageNet under 3 task settings. "B" and "C" represent the class number of the first task and the following tasks respectively and the last accuracy/average incremental accuracy are reported.

| Method | Class Number Settings | | |
|---|---|---|---|
| | B=500, C=100 | B=100, C=100 | B=10, C=10 |
| LUCIR | 49.4/57.9 | 42.3/54.8 | 21.6/30.4 |
| w/ POC | **50.8/59.0(+1.4/1.1)** | **43.8/57.3(+1.5/2.5)** | **23.2/33.6(+1.6/3.2)** |
| CwD | 50.8/58.6 | 42.8/56.2 | 22.4/31.3 |
| w/ POC | **52.4/59.8(+1.6/1.2)** | **44.8/57.6(+2.0/1.4)** | **24.5/34.0(+2.1/2.7)** |
| MEMO | 58.4/69.8 | 56.2/67.3 | 40.8/50.7 |
| w/ POC | **59.6/70.9(+1.2/1.1)** | **57.2/68.6(+1.0/1.3)** | **42.0/52.4(+1.2/1.7)** |

Table 4: Performance analysis of POC on CIFAR-100 with different sizes of memory buffer. "B", "C" and "M" represents the class number of the first task, the following tasks and the size of memory buffer for each class respectively. The last accuracy/average incremental accuracy are reported.

| Method | Class Number and Memory Buffer Size Settings | | | | | |
|---|---|---|---|---|---|---|
| | B = 50, C = 10 | | | B = 20, C = 10 | | |
| | M = 10 | M = 20 | M = 50 | M = 10 | M = 20 | M = 50 |
| LUCIR | 49.2/59.8 | 52.6/62.0 | 57.2/64.7 | 39.2/53.9 | 45.8/57.8 | 50.5/61.5 |
| w/ POC | **54.0/63.1(+4.8/3.3)** | **57.4/65.6(+4.8/3.6)** | **60.8/68.1(+3.6/3.4)** | **45.3/58.9(+6.1/5.0)** | **49.7/61.6(+3.9/3.8)** | **55.1/65.4(+4.6/3.9)** |
| CwD | 54.4/64.5 | 58.2/66.7 | 63.0/70.0 | 45.7/58.2 | 51.2/62.5 | 57.6/67.0 |
| w/ POC | **57.5/67.0(+3.1/2.5)** | **61.1/69.6(+2.9/2.9)** | **65.7/72.6(+2.7/2.6)** | **51.6/64.0(+5.9/5.8)** | **55.6/67.1(+4.4/4.6)** | **61.9/71.2(+4.3/4.2)** |
| PODNet | 43.5/56.6 | 48.7/60.3 | 56.0/64.9 | 32.6/49.3 | 38.7/53.9 | 46.5/59.4 |
| w/ POC | **49.2/62.4(+5.7/5.8)** | **53.1/64.3(+4.4/4.0)** | **61.2/69.6(+5.2/4.7)** | **36.7/54.0(+4.1/4.7)** | **42.6/58.4(+3.9/4.5)** | **50.9/63.9(+4.4/4.5)** |
| MEMO | 56.8/63.9 | 60.1/66.0 | 64.4/70.7 | 52.4/63.2 | 58.7/67.7 | 62.3/70.6 |
| w/ POC | **58.7/65.5(+1.9/1.6)** | **61.2/67.2(+1.1/1.2)** | **65.9/71.5(+1.5/0.8)** | **54.9/65.3(+2.5/2.1)** | **60.0/69.1(+1.3/1.4)** | **63.6/71.8(+1.3/1.2)** |
| LODE | 52.2/64.5 | 57.3/66.3 | 61.6/69.7 | 48.2/61.6 | 53.5/65.7 | 59.2/69.6 |
| w/ POC | **53.3/65.9(+1.1/1.4)** | **59.1/67.4(+1.8/1.1)** | **62.8/70.8(+1.2/1.1)** | **49.8/64.8(+1.6/3.2)** | **57.0/69.0(+3.5/3.3)** | **63.7/73.7(+4.5/4.1)** |
| MRFA | 54.6/64.8 | 58.2/66.5 | 63.1/70.2 | 50.4/62.4 | 55.3/66.5 | 60.7/70.2 |
| w/ POC | **55.9/66.5(+1.3/1.7)** | **59.8/67.9(+1.6/1.4)** | **64.4/71.5(+1.3/1.3)** | **52.4/64.8(+2.0/2.4)** | **57.9/68.5(+2.6/2.0)** | **63.9/72.9(+3.2/2.7)** |

(3)10, 10. The models will be trained for 160 epochs on CIFAR-100 while the training epoch will be 100 on ImageNet-100 and ImageNet.

## 4.2 EXPERIMENTAL RESULTS

### 4.2.1 COMPARISON WITH STATE-OF-THE-ARTS

We first compare the performance of LUCIR, CwD, PODNet, MEMO, LODE and MRFA before and after adopting POC to prove its effectiveness. The experiments on CIFAR-100 will run for 3 times with the random seed being 0, 42, 1993 and the mean and variance of the results will be reported. Employing a memory buffer size of 20 for each class, results are detailed in Tables 1, 2, and 3. Across all task and dataset settings, POC consistently enhances the performance of these six CIL methods.

We then evaluate the performance of the methods on CIFAR-100 under task settings (1) and (4), with different sizes of memory buffer that are 10, 20, 50 for each class. To accelerate the experiments, the random seed is fixed to be 1993 and the training epoch is set to be 100. The results in Table 4 reveal that POC also helps improve the performance of CIL methods with different buffer sizes.

We note that LUCIR, PODNet, LODE and MRFA are replay-based methods. CwD is a regularization-based method and MEMO is an architecture-based method. The improved performance of all methods with POC proves that our POC can be adapted by different categories of methods, showing the effectiveness and universality of our POC framework.

### 4.2.2 COMPARISON WITH SIMILAR WORK

In Section 2.1, we analyze the difference between our POC and FACT (Zhou et al., 2022), IL2A (Zhu et al., 2021). We point out that predicting the future classes with the seen classes through mixup is difficult. In this section, we design experiments to show that. First, we define $Overlap$ between two

Table 5: Comparison between IL2A and POC to show that POC performs better and prove that their principles are different. "B" and "C" represent the class number of the first and the following tasks and the last accuracy/average incremental accuracy are reported.

| Method | Class Number Settings | | | | | |
| | B = 50 | | | B = 20 | | |
| | C = 10 | C = 5 | C = 1 | C = 10 | C = 5 | C = 1 |
|---|---|---|---|---|---|---|
| LUCIR | 52.6/62.0 | 50.1/59.9 | 45.2/55.9 | 45.8/57.8 | 42.4/54.5 | 36.4/48.3 |
| w/ IL2A | 56.3/64.4 | 53.2/61.5 | 48.6/57.8 | 48.6/60.2 | 45.3/55.9 | 39.4/50.6 |
| w/ POC | 57.4/65.6 | 54.6/62.3 | 50.9/59.7 | 49.7/61.6 | 46.6/57.4 | 42.0/53.1 |
| w/ POC+IL2A | **58.5/66.7** | **56.3/63.4** | **52.2/61.4** | **50.8/62.7** | **47.7/58.2** | **43.1/54.3** |

Table 6: Ablation study on CIFAR-100 to show that the deterministic contrastive loss (DCL) helps improve the performance of POC. "B" and "C" represent the class number of the first and the following tasks and the last accuracy/average incremental accuracy are reported.

| Method | Class Number Settings | | | | | |
| | B = 50 | | | B = 20 | | |
| | C = 10 | C = 5 | C = 1 | C = 10 | C = 5 | C = 1 |
|---|---|---|---|---|---|---|
| LUCIR | 52.6/62.0 | 50.1/59.9 | 45.2/55.9 | 45.8/57.8 | 42.4/54.5 | 36.4/48.3 |
| w/ POC (no DCL) | 56.5/64.7 | 52.8/61.2 | 49.6/57.6 | 48.7/60.3 | 45.2/56.0 | 40.0/51.1 |
| w/ POC | **57.4/65.6** | **54.6/62.3** | **50.9/59.7** | **49.7/61.6** | **46.6/57.4** | **42.0/53.1** |
| CwD | 58.2/66.7 | 53.7/62.7 | 50.7/59.9 | 51.2/62.5 | 47.9/59.4 | 42.9/53.8 |
| w/ POC (no DCL) | 60.2/68.2 | 54.1/63.8 | 51.8/61.2 | 53.9/65.4 | 50.7/62.4 | 44.9/56.7 |
| w/ POC | **61.1/69.6** | **54.9/64.6** | **52.9/62.4** | **55.6/67.1** | **52.5/64.2** | **46.8/59.1** |
| MEMO | 60.1/66.0 | 60.6/65.9 | 56.8/61.8 | 58.7/67.7 | 59.2/67.8 | 55.2/63.5 |
| w/ POC (no DCL) | 60.8/66.8 | 61.2/66.4 | 57.4/62.3 | 59.5/68.4 | 60.1/68.2 | 56.4/64.3 |
| w/ POC | **61.2/67.2** | **61.9/67.4** | **58.5/63.8** | **60.0/69.1** | **60.8/68.9** | **57.3/65.1** |

distributions $C_i, C_j$:

$$d_{\text{intra}}(C_i) = \frac{1}{N_i} \sum_j^{N_i} d(x_j^i, \frac{1}{N_i} \sum_k^{N_i} x_k^i), d_{\text{inter}}(C_i, C_j) = d(\frac{1}{N_i} \sum_k^{N_i} x_k^i, \frac{1}{N_j} \sum_l^{N_j} x_l^j)$$

$$Overlap(C_i, C_j) = d_{\text{inter}}(C_i, C_j) - (d_{\text{intra}}(C_i) + d_{\text{intra}}(C_j)), \tag{10}$$

where $x_j^i$ is the $j$-th sample from distribution $C_i$. $d$ is the Euclidean distance. The higher the $Overlap(C_i, C_j)$ is, the more separate the distribution $C_i$ and $C_j$ are. We train a ResNet-32 on CIFAR-100 under task setting (1) and calculate the average $Overlap$ between the newly introduced classes and the mixup results of the seen classes. The values of $Overlap$ during the whole training procedure are 0.23, 0.35, 0.37, 0.28, 0.25 after normalization. These positive values indicate that the newly introduced classes and the mixup results of the seen classes are separate.

We then compare the performance of POC and IL2A to show that our POC helps the CIL methods obtain better performance. To accelerate the experiments, the random seed is fixed to be 1993 and the training epoch is set to be 100. We adopt LUCIR with IL2A and POC respectively and the results in Table 5 show that POC performs better than IL2A. Moreover, when adding IL2A and POC to LUCIR simultaneously, the performance could further increase, proving the different principle of IL2A and POC. Since FACT is not a plug-and-play method, we do not compare with LUCIR+FACT.

### 4.2.3 EFFECT OF DETERMINISTIC CONTRASTIVE LOSS

In this section, we assess POC's performance on CIFAR-100 without deterministic contrastive loss (DCL), with random seed and training epoch to be 1993 and 100. The results in Table 6 reveal that excluding DCL leads to performance degradation across all 6 task settings, highlighting its necessity.

Furthermore, our analysis of ICD, ICG in Figure 3 under task settings (1) and (4) demonstrates that ICD will be high without POC, which quantitatively shows the over-collapse and overlapping problem. Regardless of DCL, POC consistently lowers ICD, preventing future and seen classes from overlapping in model's feature space. However, without DCL, a decline in ICG indicates the compromised model generalization on seen classes, showing DCL's role in preserving it.

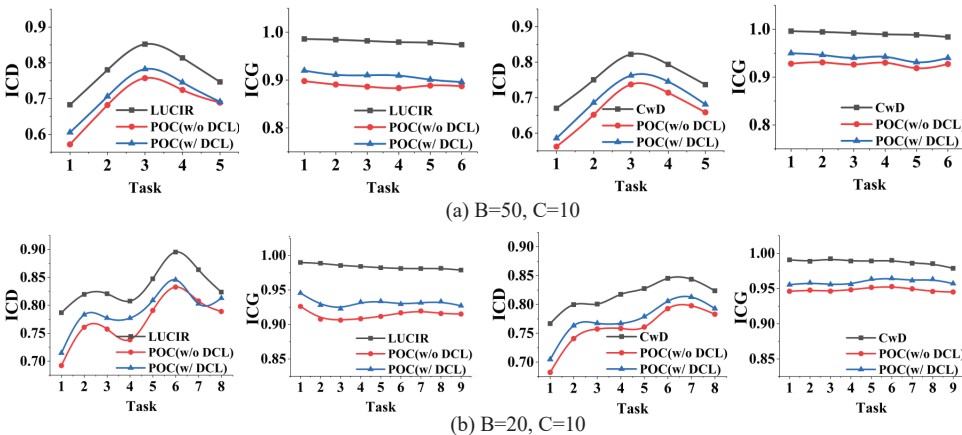

Figure 3: Illustration to show that POC protects the future classes from overlapping. The deterministic contrastive loss (DCL) protects the generalization on the seen classes.

Table 7: Analysis on CIFAR-100 showcasing the enhanced performance of POC with learnable transformation parameters. "B" and "C" represent the class number of the first task and the following tasks respectively. The last/average incremental accuracy are reported.

| Method | Class Number Settings | | | |
|---|---|---|---|---|
| | B = 50 | | B = 20 | |
| | C = 10 | C = 1 | C = 10 | C = 1 |
| LUCIR | 52.6/62.0 | 45.2/55.9 | 45.8/57.8 | 36.4/48.3 |
| Fixed | 56.5/64.8 | 49.7/57.6 | 49.0/60.2 | 40.5/52.3 |
| Learnable | **57.4/65.6** | **50.9/59.7** | **49.7/61.6** | **42.0/53.1** |

### 4.2.4 EFFECT OF LEARNABLE TRANSFORMATION

In Section 3.2.1, we emphasize the efficacy of learnable parameters $\{\theta_i\}_{i=1}^n$ by making the transformations in POC adaptable and similar to rotations. This section conducts an evaluation when employing fixed parameters as rotations and setting random seed/training epoch to be 1993/100:

$$\{\theta_i\}_{i=1}^n = \left\{ \begin{pmatrix} \cos \frac{2\pi i}{n} & -\sin \frac{2\pi i}{n} & 0 \\ \sin \frac{2\pi i}{n} & \cos \frac{2\pi i}{n} & 0 \end{pmatrix} \right\}_{i=1}^n . \tag{11}$$

The results with LUCIR as the baseline are in Table 7 and more results are supplemented in Section C.1. The results underscore the enhanced performance of POC when employing learnable transformations. Part of the rational behind this phenomenon has been stated in Section 3.2.1. In addition, with the help of modified classification loss, transformations that can generate samples without overlapping with the original samples while close to the original samples will be learned, increasing the effectiveness of POC. As a proof, we calculate the average $Overlap$ as in Section 4.2.2 between seen class and one of its transformed results. After learning 5 classes on CIFAR-100, the $Overlap$ increases from -0.17 to 0.06 when changing the direct rotations to learnable transformations, indicating that the transformed results and seen class changes from overlapped to separated while the transformed results are still close to the seen class so that representative of their adjacent regions.

## 5 CONCLUSION

In this paper, we analyze the over-collapse problem and propose a framework to Prevent the Over-Collapse (POC) for CIL. POC learns and applies a set of transformations to the seen classes to produce samples in adjacent regions. Then the model's optimization objective is modified to classify between the adjacent regions and the seen classes to prevent the over-collapse. We also introduce a deterministic contrastive loss to preserve the model's generalization on the seen classes. Extensive experiments show the POC's effectiveness in improving the performance of existing CIL methods.

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

# A PROOFS

## A.1 PROOF OF PROPOSITION 3.1

We assume that the probability distribution of $\{x\}$ corresponds to a multi-variant Gaussian distribution. Then the concrete formulation of $P_r(x)$ is:

$$P_r(x) = \frac{1}{(2\pi)^{\frac{n}{2}}|\Sigma|^{\frac{1}{2}}}e^{-\frac{(x-\mu)^T\Sigma^{-1}(x-\mu)}{2}}, \tag{12}$$

where $\mu$ and $\Sigma$ are the expectation and variance. $n$ is the dimension of $x$.

We reversely prove the following proposition, which is equal to **Proposition 3.1**:

**Proposition A.1.** *For $\forall 0 < \beta < \max(P_r(x)), 0 < \delta < 1, \exists A_a \in \{A|A \text{ is the matrix of one affine transformation}\}$, s.t. $P(P_r(A_ax) \geq \beta) \leq 1 - \delta$.*

In order to prove the **Proposition A.1**, we first prove the following proposition:

**Proposition A.2.** *For a reversable matrix $A$, if $\lambda_{min}(A^T\Sigma^{-1}A) \geq \frac{-8\log((2\pi)^{\frac{n}{2}}|\Sigma|^{\frac{1}{2}}\beta)}{2\pi|\Sigma|^{\frac{1}{n}}\sqrt[n]{(1-\delta)^2}}$, then $P(P_r(Ax) \geq \beta) \leq 1 - \delta$. $\lambda_{min}$ returns the minimum eigenvalue of the input matrix.*

*Proof.* For a reversable matrix $A$, we have the following equation:

$$P(P_r(Ax) \geq \beta) = P(\frac{1}{(2\pi)^{\frac{n}{2}}|\Sigma|^{\frac{1}{2}}}e^{-\frac{(Ax-\mu)^T\Sigma^{-1}(Ax-\mu)}{2}} \geq \beta) \tag{13}$$

$$= P(e^{-\frac{(Ax-\mu)^T\Sigma^{-1}(Ax-\mu)}{2}} \geq (2\pi)^{\frac{n}{2}}|\Sigma|^{\frac{1}{2}}\beta) \tag{14}$$

$$= P(-\frac{(Ax-\mu)^T\Sigma^{-1}(Ax-\mu)}{2} \geq \log((2\pi)^{\frac{n}{2}}|\Sigma|^{\frac{1}{2}}\beta)) \tag{15}$$

$$= P((Ax-\mu)^T\Sigma^{-1}(Ax-\mu) \leq -2\log((2\pi)^{\frac{n}{2}}|\Sigma|^{\frac{1}{2}}\beta)) \tag{16}$$

$$= P((x-A^{-1}\mu)^TA^T\Sigma^{-1}A(x-A^{-1}\mu) \leq -2\log((2\pi)^{\frac{n}{2}}|\Sigma|^{\frac{1}{2}}\beta)) \tag{17}$$

Next, we prove two lemmas and a corollary.

**Lemma A.3.** *For $\forall A \in \mathbb{R}^{n\times n}$, s.t $A^T = A$, $\forall x \in \mathbb{R}^n$, s.t $||x||_2 = 1$, we have $x^TAx \geq \lambda_{min}(A)$, where $\lambda_{min}$ returns the minimum eigenvalue of the input matrix.*

*Proof.* $\because A = A^T$ $\therefore$ we have $A = P^T\Lambda P$, where $P^TP = I$, $\Lambda$ is a diagonal matrix and the elements on the diagonal are the eigenvalues of $A$, $x^TAx = x^TP^T\Lambda Px$

Denote $Px$ as $y$, we have $||y||_2^2 = y^Ty = x^TP^TPx = x^Tx = 1$. Therefore, we have the following equation and inequation:

$$x^TAx = y^T\Lambda y \tag{18}$$

$$= \Sigma_{i=1}^n\Lambda_{ii}y_i^2 \tag{19}$$

$$\geq \min(A_{ii})\Sigma_{i=1}^ny_i^2 \tag{20}$$

$$= \lambda_{min}(A) \tag{21}$$

$\square$

**Corollary A.4.** *For $\forall A \in \mathbb{R}^{n\times n}$, s.t $A^T = A$, $\forall x \in \mathbb{R}^n$, we have $x^TAx \geq \lambda_{min}(A)||x||_2^2$, where $\lambda_{min}$ returns the minimum eigenvalue of the input matrix.*

*Proof.* We have $x^TAx = ||x||_2^2(\frac{x}{||x||_2})^TA(\frac{x}{||x||_2})$ $\because ||\frac{x}{||x||_2}||_2 = 1$. According to **Lemma** A.3, $x^TAx \geq \lambda_{min}(A)||x||_2^2$ $\square$

**Lemma A.5.** *If $\Sigma$ is a positive-definite matrix, then for $\forall A$, s.t $A$ is reversable, $A^T\Sigma A$ is a positive-definite matrix*

*Proof.* For any $x \neq \vec{0}$, $x^T A^T \Sigma A x = (Ax)^T \Sigma (Ax)$. $\because A$ is reversable $\quad \therefore Ax \neq \vec{0}$

$\because \Sigma$ is positive-definite, $Ax \neq \vec{0}$ $\quad \therefore x^T A^T \Sigma A x = (Ax)^T \Sigma (Ax) > 0$ $\quad \therefore A^T \Sigma A$ is positive-definite $\hfill \square$

Back to Equ. (15), according to **Corollary** A.4, $(x - A^{-1}\mu)^T A^T \Sigma^{-1} A(x - A^{-1}\mu) \geq \lambda_{min}(A^T\Sigma^{-1}A)||(x - A^{-1}\mu)||_2^2$

$\therefore$ If $x$ satisfies $(x - A^{-1}\mu)^T A^T \Sigma^{-1} A(x - A^{-1}\mu) \leq -2\log((2\pi)^{\frac{n}{2}}|\Sigma|^{\frac{1}{2}}\beta)$,

then $x$ must satisfies $\lambda_{min}(A^T\Sigma^{-1}A)||(x - A^{-1}\mu)||_2^2 \leq -2\log((2\pi)^{\frac{n}{2}}|\Sigma|^{\frac{1}{2}}\beta)$

$\therefore$

$$P(P_r(Ax) \geq \beta) = P((x - A^{-1}\mu)^T A^T \Sigma^{-1} A(x - A^{-1}\mu) \leq -2\log((2\pi)^{\frac{n}{2}}|\Sigma|^{\frac{1}{2}}\beta)) \tag{22}$$

$$\leq P(\lambda_{min}(A^T\Sigma^{-1}A)||x - A^{-1}\mu||_2^2 \leq -2\log((2\pi)^{\frac{n}{2}}|\Sigma|^{\frac{1}{2}}\beta)) \tag{23}$$

$$\tag{24}$$

$\because \Sigma$ is positive-definite $\quad \therefore \Sigma^{-1}$ is positive-definite. According to **Lemma** A.5, $A^T\Sigma^{-1}A$ is positive-definite

$\therefore \lambda_{min}(A^T\Sigma^{-1}A) > 0$ and we can have the following equation and inequation:

$$P(P_r(Ax) \geq \beta) \leq P(\lambda_{min}(A^T\Sigma^{-1}A)||x - A^{-1}\mu||_2^2 \leq -2\log((2\pi)^{\frac{n}{2}}|\Sigma|^{\frac{1}{2}}\beta)) \tag{25}$$

$$= P(||x - A^{-1}\mu||_2^2 \leq \frac{-2\log((2\pi)^{\frac{n}{2}}|\Sigma|^{\frac{1}{2}}\beta)}{\lambda_{min}(A^T\Sigma^{-1}A)}) \tag{26}$$

$$= \oint_{\zeta} P_r(x)dx \tag{27}$$

where $\zeta = \{x | ||x - A^{-1}\mu||_2^2 \leq \frac{-2\log((2\pi)^{\frac{n}{2}}|\Sigma|^{\frac{1}{2}}\beta)}{\lambda_{min}(A^T\Sigma^{-1}A)}\} \subseteq \{x | |x_i - (A^{-1}\mu)_i| \leq \sqrt{\frac{-2\log((2\pi)^{\frac{n}{2}}|\Sigma|^{\frac{1}{2}}\beta)}{\lambda_{min}(A^T\Sigma^{-1}A)}}\} = \psi$

$\therefore$

$$P(P_r(Ax) \geq \beta) \leq \oint_{\zeta} P_r(x)dx \tag{28}$$

$$\leq \frac{1}{(2\pi)^{\frac{n}{2}}|\Sigma|^{\frac{1}{2}}} \oint_{\zeta} dx \tag{29}$$

$$\leq \frac{1}{(2\pi)^{\frac{n}{2}}|\Sigma|^{\frac{1}{2}}} \oint_{\psi} dx \tag{30}$$

$$= \frac{2^n}{(2\pi)^{\frac{n}{2}}|\Sigma|^{\frac{1}{2}}} \sqrt{\frac{-2\log((2\pi)^{\frac{n}{2}}|\Sigma|^{\frac{1}{2}}\beta)}{\lambda_{min}(A^T\Sigma^{-1}A)}}^n \tag{31}$$

$$\leq 1 - \delta \quad (\because \lambda_{min}(A^T\Sigma^{-1}A) \geq \frac{-8\log((2\pi)^{\frac{n}{2}}|\Sigma|^{\frac{1}{2}}\beta}{2\pi|\Sigma|^{\frac{1}{n}}\sqrt[n]{(1-\delta)^2}}) \tag{32}$$

$\hfill \square$

According to **Proposition** A.2, if we can prove the following proposition, then **Proposition** A.1 holds.

**Proposition A.6.** $\{A|A$ *is a matrix of one affine transformation*$\} \cap \{A|\lambda_{min}(A^T\Sigma^{-1}A) \geq \frac{-8\log((2\pi)^{\frac{n}{2}}|\Sigma|^{\frac{1}{2}}\beta}{2\pi|\Sigma|^{\frac{1}{n}}\sqrt[n]{(1-\delta)^2}}\} \neq \phi$

*Proof.* $\because \Sigma$ is positive-definite $\quad \therefore \Sigma^{-1}$ is positive-definite. $\Sigma^{-1} = P^T \Lambda P$, where $P^T P = I$, $\Lambda$ is a diagonal matrix and the elements on the diagonal are the eigenvalues of $\Sigma^{-1}$.

Denote the minimum eigenvalue of $\Sigma^{-1}$ as $\lambda$ and $\frac{-8\log((2\pi)^{\frac{n}{2}}|\Sigma|^{\frac{1}{2}}\beta)}{2\pi|\Sigma|^{\frac{1}{n}}\sqrt[n]{(1-\delta)^2}}$ as $b$. If $A = P^T R P_2$, where $P_2^T P_2 = I$, $R$ is a diagonal matrix and the elements on the diagonal are the same and bigger than $\sqrt{\frac{b}{\lambda}}$, then

$$A^T \Sigma^{-1} A = P_2^T R P P^T \Lambda P P^T R P_2 \tag{33}$$

$$= P_2^T R \Lambda R P_2 \tag{34}$$

where $R\Lambda R$ is a diagonal matrix and the elements on the diagonal are bigger than b.

$$\therefore \{A | A = P^T R P_2, P_2^T P_2 = I, R = kI, k \geq \sqrt{\tfrac{b}{\lambda}}\} \subseteq \{A | \lambda_{min}(A^T \Sigma^{-1} A) \geq \frac{-8\log((2\pi)^{\frac{n}{2}}|\Sigma|^{\frac{1}{2}}\beta)}{2\pi|\Sigma|^{\frac{1}{n}}\sqrt[n]{(1-\delta)^2}}\}$$

We next prove the following proposition:

**Proposition A.7.** $\{A | A = P^T R P_2, P_2^T P_2 = I, R = kI, k \geq \sqrt{\tfrac{b}{\lambda}}\} = \{A | A^T A = kI, k \geq \tfrac{b}{\lambda}\}$

*Proof.* If $A \in \{A | A = P^T R P_2, P_2^T P_2 = I, R = kI, k \geq \sqrt{\tfrac{b}{\lambda}}\}$, then

$$A^T A = P_2^T R P P^T R P_2 \tag{35}$$

$$= P_2^T R R P_2 \tag{36}$$

$$= k^2 P_2^T P_2 \tag{37}$$

$$= k^2 I \in \{A | A^T A = kI, k \geq \tfrac{b}{\lambda}\} \quad (\because k \geq \sqrt{\tfrac{b}{\lambda}}) \tag{38}$$

If $A \in \{A | A^T A = kI, k \geq \tfrac{b}{\lambda}\}$, let $R = \sqrt{k}I$, $P_2 = R^{-1}PA$, then

$$P_2^T P_2 = A^T P^T R^{-1} R^{-1} P A \tag{39}$$

$$= \frac{1}{k} A^T P^T P A \tag{40}$$

$$= \frac{1}{k} A^T A \tag{41}$$

$$= I \tag{42}$$

$$\therefore A \in \{A | A = P^T R P_2, P_2^T P_2 = I, R = kI, k \geq \sqrt{\tfrac{b}{\lambda}}\} \qquad \square$$

Go back to **Proposition** A.6. According to **Proposition** A.7, $\{A | A^T A = kI, k \geq \tfrac{b}{\lambda}\} \subseteq \{A | \lambda_{min}(A^T \Sigma^{-1} A) \geq \frac{-8\log((2\pi)^{\frac{n}{2}}|\Sigma|^{\frac{1}{2}}\beta)}{2\pi|\Sigma|^{\frac{1}{n}}\sqrt[n]{(1-\delta)^2}}\}$. It is easy to construct a matrix $A$ of affine transformation, which satisfies $A \in \{A | A^T A = kI, k \geq \tfrac{b}{\lambda}\}$. Therefore, **Proposition** A.6 holds. Furthermore, **Proposition** A.2 and 3.1 holds. $\qquad \square$

A.2  PROOF OF PROPOSITION 3.2

We assume that the probability distribution of $\{x\}$ corresponds to a multi-variant Gaussian distribution and its expectation and variance are $\mu$ and $\Sigma$. Then the probability distribution of $\{Ax\}$ is also a multi-variant Gaussian distribution and its expectation and variance are $A\mu$ and $A^T \Sigma A$. Then we have the following equation and inequation:

$$\mathcal{W}(P(Ax), P(x)) = ||A\mu - \mu||_2^2 + Tr(A^T \Sigma A + \Sigma - 2[(A^T \Sigma A)^{\frac{1}{2}} \Sigma (A^T \Sigma A)^{\frac{1}{2}}]^{\frac{1}{2}}) \tag{43}$$

$$= ||A\mu - \mu||_2^2 + Tr(A^T \Sigma A) + Tr(\Sigma) - 2Tr([(A^T \Sigma A)^{\frac{1}{2}} \Sigma (A^T \Sigma A)^{\frac{1}{2}}]^{\frac{1}{2}}) \tag{44}$$

We next prove the following proposition and lemma:

**Proposition A.8.** *For* $\forall A$, $Tr([(A^T \Sigma A)^{\frac{1}{2}} \Sigma (A^T \Sigma A)^{\frac{1}{2}}]^{\frac{1}{2}}) \geq 0$

*Proof.* According to **Lemma** A.5, $A^T \Sigma A$ is a positive-definite matrix. Suppose that $A^T \Sigma A = P^T \Lambda P$, where $P^T P = I$, $\Lambda$ is a diagonal matrix and the elements on the diagonal are bigger than 0.

$\because (P^T \Lambda^{\frac{1}{2}} P)^2 = P^T \Lambda^{\frac{1}{2}} P P^T \Lambda^{\frac{1}{2}} P = P^T \Lambda P \quad \therefore (A^T \Sigma A)^{\frac{1}{2}} = P^T \Lambda^{\frac{1}{2}} P \quad \therefore (A^T \Sigma A)^{\frac{1}{2}} = [(A^T \Sigma A)^{\frac{1}{2}}]^T$

$\therefore (A^T \Sigma A)^{\frac{1}{2}} \Sigma (A^T \Sigma A)^{\frac{1}{2}}$ is also a positive-definite matrix.

Suppose that $(A^T \Sigma A)^{\frac{1}{2}} \Sigma (A^T \Sigma A)^{\frac{1}{2}} = P_2^T \Lambda_2 P_2$, then $[(A^T \Sigma A)^{\frac{1}{2}} \Sigma (A^T \Sigma A)^{\frac{1}{2}}]^{\frac{1}{2}} = P_2^T \Lambda_2^{\frac{1}{2}} P_2$

$$Tr([(A^T \Sigma A)^{\frac{1}{2}} \Sigma (A^T \Sigma A)^{\frac{1}{2}}]^{\frac{1}{2}}) = Tr(P_2^T \Lambda_2^{\frac{1}{2}} P_2) \tag{45}$$

$$= Tr(\Lambda_2^{\frac{1}{2}} P_2 P_2^T) \tag{46}$$

$$= Tr(\Lambda_2^{\frac{1}{2}}) \geq 0 \tag{47}$$

$\square$

**Lemma A.9.** *For $\forall A, B, s.t. A^T = A, B^T = B$, the following inequation holds: $Tr(AB) \leq \frac{1}{2}(Tr(A^2) + Tr(B^2))$*

*Proof.* $\because \forall A, B, Tr[(A+B)(A+B)^T] + Tr[(A-B)(A-B)^T] = 2[Tr(AA^T) + Tr(BB^T)]$

$\therefore \forall A, B, Tr[(A+B)(A+B)^T] \leq 2[Tr(AA^T) + Tr(BB^T)]$

When $A^T = A, B^T = B$, $Tr[(A+B)(A+B)^T] = Tr[A^2 + B^2 + AB + BA] = Tr[A^2] + Tr[B^2] + 2Tr[AB]$

$\therefore Tr(AB) \leq \frac{1}{2}(Tr(A^2) + Tr(B^2))$ $\square$

According to **Proposition** A.8 and Equ. (42), we have:

$$\mathcal{W}(P(Ax), P(x)) = ||A\mu - \mu||_2^2 + Tr(A^T \Sigma A) + Tr(\Sigma) - 2Tr([(A^T \Sigma A)^{\frac{1}{2}} \Sigma (A^T \Sigma A)^{\frac{1}{2}}]^{\frac{1}{2}}) \tag{48}$$

$$\leq ||A\mu - \mu||_2^2 + Tr(A^T \Sigma A) + Tr(\Sigma) \tag{49}$$

$$= ||A\mu - \mu||_2^2 + Tr(\Sigma AA^T) + Tr(\Sigma) \tag{50}$$

$$\leq ||A\mu - \mu||_2^2 + \frac{1}{2}(Tr(\Sigma^2) + Tr(AA^T AA^T)) + Tr(\Sigma) \quad (\because \textbf{Lemma} A.9) \tag{51}$$

$\because A$ is a matrix of rotation transformation $\quad \therefore A_{i,j} \geq 0, \forall i, \Sigma_{j=1}^n A_{i,j} = 1$

$\therefore$

$$0 \leq (AA^T)_{i,j} = \Sigma_{k=1}^n A_{i,k} A_{k,j}^T \tag{52}$$

$$\leq \sqrt{(\Sigma_{k=1}^n A_{i,k}^2)(\Sigma_{k=1}^n (A_{k,j}^T)^2)} \tag{53}$$

$$\leq \sqrt{(\Sigma_{k=1}^n A_{i,k})^2 (\Sigma_{k=1}^n A_{k,j}^T)^2} \tag{54}$$

$$= 1 \tag{55}$$

$\therefore$

$$\mathcal{W}(P(Ax), P(x)) \leq ||A\mu - \mu||_2^2 + \frac{1}{2}(Tr(\Sigma^2) + Tr(AA^TAA^T)) + Tr(\Sigma) \tag{56}$$

$$\leq ||A\mu - \mu||_2^2 + \frac{1}{2}(Tr(\Sigma^2) + n^2) + Tr(\Sigma) \quad (\because 0 \leq (AA^TAA^T)_{i,j} \leq n) \tag{57}$$

$$\leq ||A\mu||_2^2 + ||\mu||_2^2 + \frac{1}{2}(Tr(\Sigma^2) + n^2) + Tr(\Sigma) \tag{58}$$

$$= \Sigma_{i=1}^n (\Sigma_{j=1}^n A_{i,j}\mu_j)^2 + ||\mu||_2^2 + \frac{1}{2}(Tr(\Sigma^2) + n^2) + Tr(\Sigma) \tag{59}$$

$$\leq \Sigma_{i=1}^n ((\Sigma_{j=1}^n A_{i,j}^2)(\Sigma_{j=1}^n \mu_j^2)) + ||\mu||_2^2 + \frac{1}{2}(Tr(\Sigma^2) + n^2) + Tr(\Sigma) \tag{60}$$

$$\leq \Sigma_{i=1}^n ||\mu||_2^2 + ||\mu||_2^2 + \frac{1}{2}(Tr(\Sigma^2) + n^2) + Tr(\Sigma) \quad (\because A_{i,j} \geq 0, \forall i, \Sigma_{j=1}^n A_{i,j} = 1) \tag{61}$$

Inequation (51) and (58) hold because of Cauchy inequation. Therefore, **Proposition** 3.2 holds.

## B  DETAILS OF MODELS AND TRAINING

We employ ResNet-32 for CIFAR-100 and ResNet-18 for ImageNet-100, ImageNet as our classification models, adhering to the conventional methodology. These models process inputs through a sequence of convolutional layers to extract features, subsequently reshaping them into vectors using global max-pooling. The resultant vector serves as input for a fully connected network, responsible for calculating probabilities across seen classes. This setup defines the convolutional layers with max-pooling as the feature extractor $f$ and the fully connected network as the classifier $\Phi$ used in determining deterministic contrastive loss. For CIFAR-100 and ImageNet-100 under task settings (3) and (6) of the main manuscript, to mitigate overfitting on the subsequent tasks, we reset the training epochs to 20 and the learning rate to 0.01 at the beginning of task 2. The training batch size on CIFAR-100, ImageNet-100 and ImageNet are 128, 32 and 32 respectively. All experiments are finished on GeForce RTX 3090 GPUs. On CIFAR-100, ImageNet-100 and ImageNet, one run of experiment needs 8 hours, 5 days and 15 days respectively.

## C  COMPLEMENTARY EXPERIMENTS

This section supplements additional experiments not featured in the main manuscript. To accelerate the experiments, in all experiments, the random seed is fixed to be 1993 and the training epoch is set to be 100. In Section C.1, we assess POC's performance when fixing transformation parameters to be rotations, emphasizing the benefits of learnable parameters. In Section C.2, we investigate whether the performance increases when the produced samples are regarded as positive and have the same labels as their original classes. Section C.3 analyze the performance of POC using different transformation types. Section C.4 presents experiments where adjacent regions are solely constrained to be close to original classes, affirming the importance of maintaining diverse transformations. Section C.5 showcases experiments where adjacent regions receive identical labels, highlighting the superior performance achieved by assigning distinct labels. In Section C.6, we explore how varying the number of transformations impacts POC's performance. In Section C.7, we integrate our POC into the models for other incremental learning tasks to show that POC is compatible and generalizable to other tasks. Section C.8 investigates the sensitivity of POC's performance to hyperparameters $\lambda_1$ and $\lambda_2$. Lastly, Section C.9 discusses the training costs of POC.

### C.1  EFFECT OF LEARNABLE TRANSFORMATION

In Section 3.2.1, we emphasize the efficacy of learnable parameters $\{\theta_i\}_{i=1}^n$ by making the transformations in POC adaptable and similar to rotations. And in Section 4.2.4, the performance of POC with LUCIR as the baseline is evaluated when the transformations are fixed to be rotations. Following

Table 8: Analysis on CIFAR-100 showcasing the enhanced performance of POC with learnable transformation parameters. "B" and "C" represent the class number of the first task and the following tasks respectively. The last/average incremental accuracy are reported.

| Method | Class Number Settings | | | | | |
| --- | --- | --- | --- | --- | --- | --- |
| | B = 50 | | | B = 20 | | |
| | C = 10 | C = 5 | C = 1 | C = 10 | C = 5 | C = 1 |
| CwD | 58.2/66.7 | 53.7/62.7 | 50.7/59.9 | 51.2/62.5 | 47.9/59.4 | 42.9/53.8 |
| Fixed | 59.6/68.3 | 54.1/63.6 | 51.6/61.2 | 53.9/65.2 | 50.3/62.5 | 45.1/56.4 |
| Learnable | **61.1/69.6** | **54.9/64.6** | **52.9/62.4** | **55.6/67.1** | **52.5/64.2** | **46.8/59.1** |
| PODNet | 48.7/60.3 | 48.9/59.8 | 49.3/59.9 | 38.7/53.9 | 36.1/51.0 | 36.1/50.3 |
| Fixed | 51.8/62.4 | 51.7/62.1 | 50.9/61.6 | 41.2/57.0 | 39.5/54.8 | 37.8/51.4 |
| Learnable | **53.1/64.3** | **53.3/63.8** | **52.3/63.3** | **42.6/58.4** | **41.0/56.5** | **39.5/53.6** |
| MEMO | 60.1/66.0 | 60.6/65.9 | 56.8/61.8 | 58.7/67.7 | 59.2/67.8 | 55.2/63.5 |
| Fixed | 60.6/66.5 | 61.2/66.4 | 57.6/62.9 | 59.4/68.2 | 59.9/68.4 | 56.2/64.3 |
| Learnable | **61.2/67.2** | **61.9/67.4** | **58.5/63.8** | **60.0/69.1** | **60.8/68.9** | **57.3/65.1** |
| LODE | 57.3/66.3 | 51.3/60.6 | 49.2/58.7 | 53.5/65.7 | 50.0/63.5 | 48.5/59.3 |
| Fixed | 58.3/66.9 | 52.4/61.2 | 50.2/59.6 | 55.6/67.3 | 51.0/64.0 | 49.4/60.5 |
| Learnable | **59.1/67.4** | **53.8/62.0** | **51.3/60.4** | **57.0/69.0** | **52.1/64.6** | **50.2/62.1** |
| MRFA | 58.2/66.5 | 55.7/63.4 | 52.4/60.5 | 55.3/66.5 | 53.2/64.6 | 51.1/61.2 |
| Fixed | 59.0/67.1 | 56.4/64.5 | 53.3/61.7 | 56.8/67.8 | 54.9/65.6 | 52.5/63.0 |
| Learnable | **59.8/67.9** | **57.3/65.9** | **54.7/62.6** | **57.9/68.5** | **56.3/66.5** | **53.7/64.1** |

Table 9: Analysis on ImageNet-100 showcasing the enhanced performance of POC with learnable transformation parameters. "B" and "C" represent the class number of the first task and the following tasks respectively. The last/average incremental accuracy are reported.

| Method | Class Number Settings | | | | | |
| --- | --- | --- | --- | --- | --- | --- |
| | B = 50 | | | B = 20 | | |
| | C = 10 | C = 5 | C = 1 | C = 10 | C = 5 | C = 1 |
| CwD | 60.4/71.6 | 55.8/68.2 | 40.3/56.3 | 48.2/62.9 | 44.6/58.5 | 34.3/51.1 |
| Fixed | 61.2/72.4 | 56.6/68.9 | 42.9/58.2 | 50.0/63.8 | 45.9/59.6 | 37.3/52.0 |
| Learnable | **62.3/73.2** | **57.4/69.4** | **44.7/59.8** | **51.2/64.4** | **47.1/60.6** | **38.9/53.1** |
| PODNet | 62.3/73.4 | 57.4/71.6 | 42.9/59.7 | 45.8/63.0 | 41.7/59.8 | 32.4/50.0 |
| Fixed | 63.1/74.2 | 60.7/72.2 | 46.8/62.1 | 47.4/64.0 | 46.6/61.1 | 35.4/53.4 |
| Learnable | **63.8/75.0** | **62.3/72.8** | **48.6/63.7** | **49.1/64.8** | **48.2/62.1** | **36.6/55.1** |
| MEMO | 66.2/76.8 | 64.5/76.4 | 52.7/64.0 | 53.6/67.1 | 48.4/60.8 | 40.3/53.2 |
| Fixed | 66.7/77.3 | 65.7/77.1 | 54.5/65.3 | 54.6/67.9 | 49.6/61.7 | 41.5/54.0 |
| Learnable | **67.4/77.9** | **66.5/77.8** | **55.9/66.5** | **55.4/68.2** | **50.7/62.5** | **42.4/54.7** |
| LODE | 64.5/73.6 | 59.4/71.0 | 45.8/60.4 | 50.6/63.5 | 45.3/59.5 | 37.2/52.1 |
| Fixed | 65.3/74.6 | 60.6/72.3 | 48.8/62.5 | 52.4/64.8 | 46.9/60.7 | 39.2/52.8 |
| Learnable | **66.1/75.1** | **61.7/73.1** | **50.4/63.8** | **53.4/65.7** | **48.3/62.3** | **40.5/53.4** |
| MRFA | 65.1/74.8 | 61.4/73.2 | 47.3/61.6 | 51.8/64.9 | 46.1/60.0 | 38.5/52.6 |
| Fixed | 65.8/75.2 | 62.3/74.0 | 48.5/63.1 | 52.7/65.7 | 46.9/60.9 | 39.4/53.4 |
| Learnable | **66.4/76.0** | **63.3/74.9** | **50.6/64.4** | **54.1/66.6** | **48.7/62.1** | **41.1/54.3** |

the setting of Section 4.2.4, this section conducts an evaluation of performance when changing the baseline to CwD, PODNet, MEMO, LODE and MRFA to further support our motivation.

The results in Table 8, 9 and 10 underscore the enhanced performance of POC when employing learnable transformations. According to the results, when the transformations are fixed, the last/average incremental accuracy of baseline methods will increase 1.4/1.3, 1.6/1.1, 0.9/1.1 on CIFAR-100, ImageNet-100 and ImageNet in average respectively. However, when setting the transformations learnable, the last/average incremental accuracy of baseline methods will increase 2.6/2.6, 2.9/2.1, 1.7/1.8 in average on CIFAR-100, ImageNet-100 and ImageNet. It shows that the performance

Table 10: Analysis on ImageNet showcasing the enhanced performance of POC with learnable transformation parameters. "B" and "C" represent the class number of the first task and the following tasks respectively. The last/average incremental accuracy are reported.

| Method | Class Number Settings | | |
|---|---|---|---|
| | B=500, C=100 | B=100, C=100 | B=10, C=10 |
| LUCIR | 49.4/57.9 | 42.3/54.8 | 21.6/30.4 |
| Fixed | 50.3/58.6 | 43.2/56.2 | 22.6/32.4 |
| Learnable | **50.8/59.0** | **43.8/57.3** | **23.2/33.6** |
| CwD | 50.8/58.6 | 42.8/56.2 | 22.4/31.3 |
| Fixed | 51.8/59.4 | 44.1/57.1 | 23.6/32.8 |
| Learnable | **52.4/59.8** | **44.8/57.6** | **24.5/34.0** |
| MEMO | 58.4/69.8 | 56.2/67.3 | 40.8/50.7 |
| Fixed | 59.1/70.5 | 56.9/68.2 | 41.5/51.8 |
| Learnable | **59.6/70.9** | **57.2/68.6** | **42.0/52.4** |

gain will increase by 185%/200%, 181%/190%, 188%/163% when setting transformations learnable, showing the efficacy of learnable parameters. Further to the explanation in Section 4.2.4, the rationale behind lies in the adaptability afforded by learnable parameters. By enabling learnable transformations, the loss optimization process gains flexibility in shaping the feature space and transforming parameters. Consequently, the model learns transformations capable of generating more representative adjacent regions of seen classes in the input space, resulting in improved performance.

## C.2    EFFECT OF SAMPLE LABEL SETTING

In the original POC, the samples produced by learnable transformations are considered as negative samples of seen classes and have different labels. It is possible that the performance improves because the model has seen multiple augmented samples. Therefore, we report and compare the performance when the produced samples are considered as positive samples and have the same labels as their original classes. Following the setting of Section 4.2.1, we report the performance of different baselines when the produced samples are considered positive. The results in Table 11 and 12 show that the performance gain is lower than that when the produced samples are regarded as negative.

We further analyze the Inter-Class Distance (ICD) and Intra-Class Generalization (ICG) when the produced samples are considered as positive samples. The results illustrated in Figure 4 indicates that when the produced samples are considered positive, both ICG and ICD will increase. This results from that when produced samples are considered positive, since the rotated samples have similar distribution with original ones as shown in Section 4.2.4 and the transform loss $\mathcal{L}_{\text{Trans}}$ makes the transformations similar to rotations, the transformations will converge to rotations to minimize both $\mathcal{L}_{\text{Mod\_Cls}}$ and $\mathcal{L}_{\text{Trans}}$ so that $\mathcal{L}_{\text{Total}}$ is minimized. Therefore, the situation will be the same as that when training with rotation augmentation so that the decision boundary of one class will extend more broadly. Although it can increase the model's generalization, the over-collapse is more severe as well so that the overlapping between seen and future classes is worsen. Under the combined effect, the performance gain is minor. Instead, the original design of POC both prevents over-collapse and protects the generalization so that it obtains better performance.

## C.3    EFFECT OF TRANSFORMATION TYPE

As outlined in Section 3.2.1, we confine the transformations in POC to be similar to rotations. In this section, using LUCIR as the baseline method and CIFAR-100 for evaluation, we make the transformations resemble other alternatives, including identical, mixup, cutmix, flip, Gaussian blur and Gaussian noise, to assess their efficacy. The results in Table 13 demonstrate varying performance among different transformations, with rotation notably enhancing POC's performance.

We also conduct a comparative analysis of Inter-Class Distance (ICD) and Intra-Class Generalization (ICG) among diverse transformation types, visualized in Figure 5. The findings reveal that rotations strike a better balance between ICD and ICG, resulting in enhanced overall performance for POC.

Table 11: Analysis on CIFAR-100 to show that regarding samples produced by learnable transformations as negative to their original classes helps POC obtain better performance. "B" and "C" represent the class number of the first task and the following tasks respectively. The experiments are run for 3 times and the mean and variance of average incremental accuracy are reported.

| Method | Class Number Settings | | | | | |
| | B = 50 | | | B = 20 | | |
| | C = 10 | C = 5 | C = 1 | C = 10 | C = 5 | C = 1 |
|---|---|---|---|---|---|---|
| LUCIR | 64.1±0.9 | 61.2±0.7 | 55.9±0.3 | 59.4±0.5 | 57.6±0.3 | 48.5±0.2 |
| Positive | 64.9±1.0 | 61.8±0.8 | 56.7±0.3 | 60.5±0.7 | 58.2±0.2 | 49.6±0.5 |
| Negative | **66.8±0.7** | **63.5±0.6** | **59.6±0.4** | **63.8±0.3** | **59.2±0.3** | **53.1±0.2** |
| CwD | 67.2±0.2 | 62.8±0.1 | 59.7±0.2 | 64.3±0.4 | 61.2±0.5 | 53.6±0.3 |
| Positive | 68.0±0.4 | 63.8±0.1 | 60.6±0.4 | 65.1±0.5 | 62.1±0.3 | 54.7±0.4 |
| Negative | **69.6±0.4** | **65.4±0.3** | **62.3±0.5** | **68.3±0.2** | **66.1±0.4** | **59.1±0.2** |
| PODNet | 64.6±0.7 | 63.2±1.1 | 59.8±0.5 | 54.9±0.4 | 53.2±0.4 | 50.5±0.2 |
| Positive | 65.2±0.6 | 63.7±1.1 | 60.5±0.7 | 56.0±0.6 | 54.0±0.6 | 51.2±0.4 |
| Negative | **68.2±0.8** | **67.2±1.0** | **63.1±0.7** | **60.6±0.7** | **58.3±0.4** | **53.5±0.5** |
| MEMO | 70.2±0.5 | 69.0±0.7 | 61.4±0.3 | 69.5±0.5 | 67.3±0.8 | 63.2±0.4 |
| Positive | 70.8±0.7 | 69.6±0.6 | 62.0±0.5 | 70.0±0.5 | 67.9±0.5 | 63.7±0.4 |
| Negative | **71.8±0.6** | **70.4±0.4** | **63.5±0.5** | **70.9±0.6** | **69.3±0.4** | **64.8±0.6** |
| LODE | 68.7±0.6 | 64.6±0.8 | 58.5±0.4 | 66.2±0.5 | 64.4±0.3 | 59.2±0.5 |
| Positive | 69.2±0.7 | 65.1±0.5 | 58.9±0.5 | 67.0±0.6 | 64.8±0.4 | 59.7±0.6 |
| Negative | **70.0±0.5** | **66.1±0.7** | **60.5±0.7** | **68.4±0.3** | **65.8±0.6** | **62.4±0.4** |
| MRFA | 68.0±0.4 | 66.4±0.6 | 60.3±0.8 | 67.8±0.8 | 65.7±0.6 | 61.3±0.7 |
| Positive | 68.5±0.5 | 66.8±0.5 | 60.8±0.7 | 68.4±0.7 | 66.2±0.6 | 62.0±0.5 |
| Negative | **69.2±0.4** | **68.1±0.5** | **62.7±0.5** | **69.6±0.6** | **67.5±0.4** | **63.6±0.8** |

Table 12: Analysis on ImageNet-100 to show that regarding samples produced by learnable transformations as negative to their original classes helps POC obtain better performance. "B" and "C" represent the class number of the first task and the following tasks respectively and the last accuracy/average incremental accuracy are reported.

| Method | Class Number Settings | | | | | |
| | B = 50 | | | B = 20 | | |
| | C = 10 | C = 5 | C = 1 | C = 10 | C = 5 | C = 1 |
|---|---|---|---|---|---|---|
| LUCIR | 61.4/71.5 | 55.1/67.2 | 41.1/56.8 | 48.0/61.5 | 42.6/55.7 | 34.3/48.9 |
| Positive | 62.1/72.3 | 55.8/67.6 | 42.1/57.7 | 48.9/62.3 | 43.4/56.6 | 34.8/49.4 |
| Negative | **64.0/73.7** | **57.6/68.3** | **47.7/61.8** | **51.5/65.2** | **46.4/59.3** | **36.9/51.5** |
| CwD | 60.4/71.6 | 55.8/68.2 | 40.3/56.3 | 48.2/62.9 | 44.6/58.5 | 34.3/51.1 |
| Positive | 60.9/72.2 | 56.3/68.6 | 40.8/56.5 | 49.0/63.7 | 45.3/59.1 | 34.9/51.7 |
| Negative | **62.3/73.2** | **57.4/69.4** | **44.7/59.8** | **51.2/64.4** | **47.1/60.6** | **38.9/53.1** |
| PODNet | 62.3/73.4 | 57.4/71.6 | 42.9/59.7 | 45.8/63.0 | 41.7/59.8 | 32.4/50.0 |
| Positive | 62.8/73.8 | 57.9/72.3 | 43.7/60.4 | 46.4/63.9 | 42.2/60.4 | 32.9/50.7 |
| Negative | **63.8/75.0** | **62.3/72.8** | **48.6/63.7** | **49.1/64.8** | **48.2/62.1** | **36.6/55.1** |
| MEMO | 66.2/76.8 | 64.5/76.4 | 52.7/64.0 | 53.6/67.1 | 48.4/60.8 | 40.3/53.2 |
| Positive | 66.7/77.3 | 65.0/76.8 | 53.4/64.6 | 54.3/67.7 | 49.1/61.7 | 40.8/53.9 |
| Negative | **67.4/77.9** | **66.5/77.8** | **55.9/66.5** | **55.4/68.2** | **50.7/62.5** | **42.4/54.7** |
| LODE | 64.5/73.6 | 59.4/71.0 | 45.8/60.4 | 50.6/63.5 | 45.3/59.5 | 37.2/52.1 |
| Positive | 64.8/74.0 | 59.7/71.5 | 46.3/60.8 | 51.0/64.0 | 45.9/59.9 | 37.8/52.6 |
| Negative | **66.1/75.1** | **61.7/73.1** | **50.4/63.8** | **53.4/65.7** | **48.3/62.3** | **40.5/53.4** |
| MRFA | 65.1/74.8 | 61.4/73.2 | 47.3/61.6 | 51.8/64.9 | 46.1/60.0 | 38.5/52.6 |
| Positive | 65.7/75.2 | 61.7/73.6 | 47.9/62.3 | 52.3/65.3 | 46.6/60.5 | 38.9/53.2 |
| Negative | **66.4/76.0** | **63.3/74.9** | **50.6/64.4** | **54.1/66.6** | **48.7/62.1** | **41.1/54.3** |

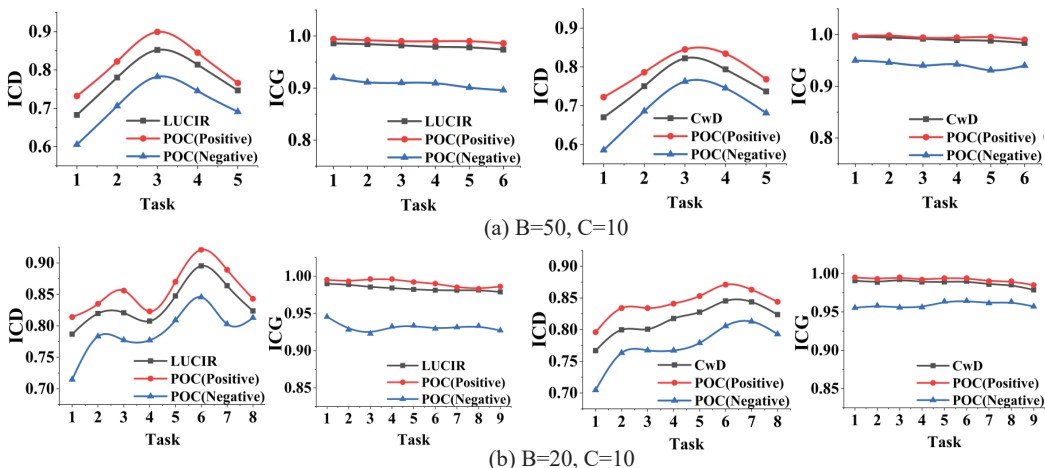

Figure 4: Illustration to show that regarding samples produced by learnable transformations as positive to their original classes increases model's generalization while overlapping is more severe.

Table 13: Analysis on CIFAR-100 to show that resembling different transformation types influences the performance a lot. "B" and "C" represent the class number of the first task and the following tasks respectively and the last accuracy/average incremental accuracy are reported.

| Transform Type | Class Number Settings | | | |
| --- | --- | --- | --- | --- |
| | B = 50 | | B = 20 | |
| | C = 10 | C = 1 | C = 10 | C = 1 |
| Baseline | 52.6/62.0 | 45.2/55.9 | 45.8/57.8 | 36.4/48.3 |
| Identical | 53.2/62.5 | 45.7/56.2 | 45.8/58.2 | 37.0/48.8 |
| Mixup | 55.8/64.2 | 48.3/57.2 | 47.4/60.0 | 40.6/51.8 |
| Cutmix | 55.3/63.7 | 47.6/56.4 | 46.9/59.3 | 40.2/51.2 |
| Flip | 53.7/63.4 | 45.6/56.2 | 46.3/58.6 | 37.0/49.2 |
| Noise | 53.5/62.8 | 46.3/56.8 | 46.7/59.0 | 37.4/50.2 |
| Blur | 52.7/62.7 | 46.0/56.3 | 46.5/58.6 | 37.2/49.7 |
| Rotation | **57.4/65.6** | **50.9/59.7** | **49.7/61.6** | **42.0/53.1** |

## C.4 EFFECT OF TRANSFORMATION DIVERSITY

In deterministic contrastive loss (DCL), the adjacent regions are kept away from each other to keep transformations diversified. This section aims to demonstrate the essentiality of maintaining this diversity. To showcase the significance of diversity preservation, we modify the DCL as follows:

$$\mathcal{L}_{\text{DCL}} = -\sum_{i=1}^{n} \text{sim}(x_i, x_0). \tag{62}$$

Initially, following the modification of the DCL, we evaluate the performance of POC using LUCIR as the baseline method, and the results on CIFAR-100 are presented in Table 14. These findings highlight that maintaining diversified transformations enhances POC performance.

Moreover, we conduct an in-depth analysis of Inter-Class Distance (ICD) and Intra-Class Generalization (ICG) during the training process. These results, depicted in Figure 6, underscore a crucial insight: while forgoing diversity may improve model generalization on seen classes, it leads to decreased distance between future and seen classes.

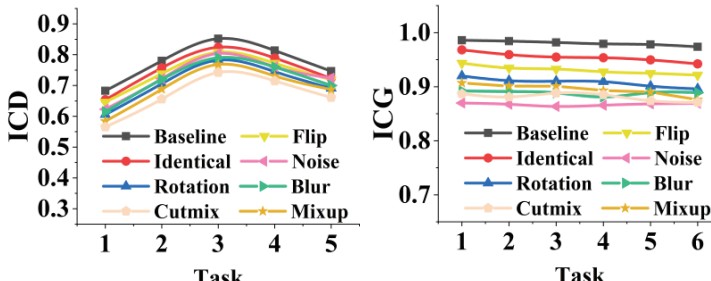

Figure 5: Illustration to show that different transformation types have different influences on the inter-class distance and intra-class generalization along the training procedure. The class number of the first task and the following tasks are 50 and 10 respectively.

Table 14: Analysis on CIFAR-100 to show the necessity of keeping the transformations diversified for POC. "B" and "C" represent the class number of the first and the following tasks and the last accuracy/average incremental accuracy are reported.

| | Class Number Settings | | | |
|---|---|---|---|---|
| Diversity | B = 50 | | B = 20 | |
| | C = 10 | C = 1 | C = 10 | C = 1 |
| Baseline | 52.6/62.0 | 45.2/55.9 | 45.8/57.8 | 36.4/48.3 |
| No | 56.7/64.9 | 50.3/58.7 | 48.9/61.2 | 40.5/51.8 |
| Keep | **57.4/65.6** | **50.9/59.7** | **49.7/61.6** | **42.0/53.1** |

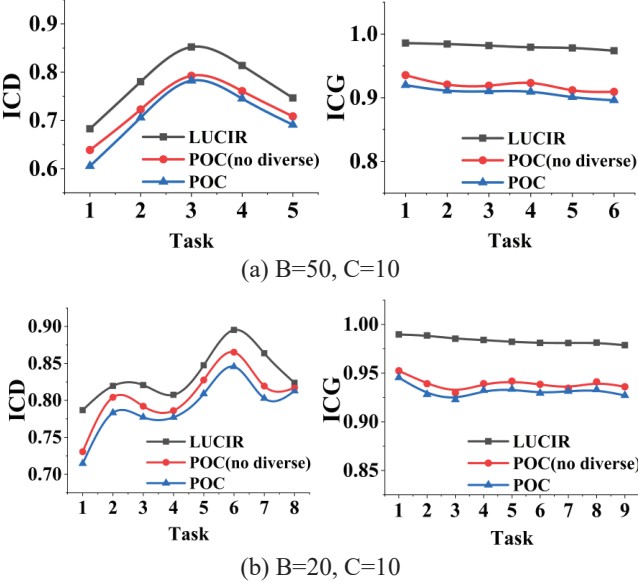

(a) B=50, C=10

(b) B=20, C=10

Figure 6: Illustration to show that keeping the transformations in POC diversified can protect the future classes from overlapping with the seen classes better.

## C.5  EFFECT OF DIFFERENT LABELS

In the main manuscript, for one seen class, the labeled adjacent regions are assigned with different labels for classification. This section assesses performance when adjacent regions labeled by one seen class are assigned with the same label. Specifically, for the labeled adjacent regions $\{\{\{(x_{i,j,k}, y_{i,k})\}_{j=1}^{N_i}\}_{i=1}^{\widetilde{L}_t}\}_{k=1}^{n}$, we set $y_{i,k}$ to equal $y_{i,l}$ and then calculates the classification loss.

Table 15: Analysis on CIFAR-100 to show that assigning the adjacent regions with different labels leads to better performance of POC. "B" and "C" represent the class number of the first task and the following tasks respectively and the last accuracy/average incremental accuracy are reported.

| Label Assign | Class Number Settings | | | |
|---|---|---|---|---|
| | B = 50 | | B = 20 | |
| | C = 10 | C = 1 | C = 10 | C = 1 |
| Baseline | 52.6/62.0 | 45.2/55.9 | 45.8/57.8 | 36.4/48.3 |
| Same | 56.7/65.2 | 48.7/57.6 | 47.8/59.8 | 38.6/48.6 |
| Different | **57.4/65.6** | **50.9/59.7** | **49.7/61.6** | **42.0/53.1** |

Table 16: Analysis on CIFAR-100 to show that transformation number has influence on the final performance. "B" and "C" represent the class number of the first task and the following tasks respectively and the last accuracy/average incremental accuracy are reported.

| Transform Number | Class Number Settings | | | |
|---|---|---|---|---|
| | B = 50 | | B = 20 | |
| | C = 10 | C = 1 | C = 10 | C = 1 |
| 1 | 55.0/63.4 | 46.3/55.0 | 47.0/58.5 | 35.0/46.8 |
| 5 | 56.7/65.1 | 50.5/59.4 | 49.5/60.7 | 39.3/50.8 |
| 7 | 56.8/64.9 | 49.5/59.2 | 48.5/60.0 | 38.7/50.9 |
| 3 | **57.4/65.6** | **50.9/59.7** | **49.7/61.6** | **42.0/53.1** |

Table 15 presents the performance of POC with LUCIR as the baseline on CIFAR-100 under task settings (1), (3), (4), and (6). The findings demonstrate that assigning different labels to adjacent regions yields superior performance.

## C.6 EFFECT OF TRANSFORMATION NUMBER

This section assesses the impact of the number of transformations under task settings (1), (3), (4), and (6), utilizing LUCIR as the baseline method. In Section 4.2.1, the number of transformations is set to be 3 and here we compare the performance of POC on CIFAR-100 when the number is 1, 5, 7. The findings, listed in Table 16, highlight that POC achieves superior performance when the transformation number is 3. This suggests that an excessively small or large number of transformations does not yield optimal results.

We further compare the Inter-Class Distance (ICD) and Intra-Class Generalization (ICG) among different transformation numbers and the results are illustrated in Figure 7. The findings indicate that while a larger number of transformations may enhance generalization on seen classes, it simultaneously diminishes the separation between future and seen classes within the feature space. This indicates the significance of selecting an optimal transformation number.

It is noteworthy that the ICD decreases as the number of transformations decreases. We analyze the reason is that when the transformation number decreases, less transformed results will push each other away due to DCL so that they will be less compact. In details, when the transformation number decreases, the transformed result of the seen class by one transformation will be affected by fewer other results through DCL, which pushes different results away from each other. Therefore, one transformed result can be less compact and can cover more adjacent regions. When the number is 1, DCL only constrains the transformed result to be close to the seen class. Therefore, the transformed result will surround the seen class to minimize the distance. In this way, training the model to classify between the seen class and more adjacent regions will decrease ICD.

To consolidate our analysis, we calculate the average cosine similarity between two samples from one transformation result of one seen class. We find that after learning 6 tasks, the average cosine similarity decreases from 0.89 to 0.8 when transformation number decreases from 7 to 1, indicating a less compact distribution of transformation results.

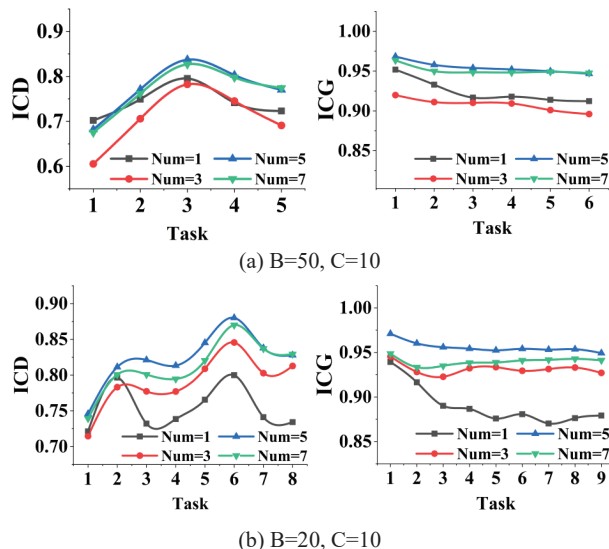

Figure 7: Illustration to show that different transformation numbers have different influences on the inter-class distance and intra-class generalization along the training procedure. "B" and "C" represent the class number of the first task and the following tasks.

Table 17: Performance analysis on PASCAL-VOC 2007 to show that POC is generalizable to incremental object detection task and the mAP is reported.

| Method | Class Number Settings | | | | | | | | | | | | | | |
| --- | --- | --- | --- | --- | --- | --- | --- | --- | --- | --- | --- | --- | --- | --- | --- |
| | 10-5 | | | 5-5 | | | 10-2 | | | 15-1 | | | 10-1 | | |
| | 1-10 | 11-20 | 1-20 | 1-5 | 6-20 | 1-20 | 1-10 | 11-20 | 1-20 | 1-15 | 16-20 | 1-20 | 1-10 | 11-20 | 1-20 |
| ABR | 68.7 | 67.1 | 67.9 | 64.7 | 56.4 | 58.4 | 67.0 | 58.1 | 62.6 | 68.7 | 56.7 | 65.7 | 62.0 | 55.7 | 58.9 |
| w/ POC | **69.8** | **68.6** | **69.2** | **66.2** | **58.6** | **60.5** | **68.1** | **59.4** | **63.8** | **69.6** | **58.4** | **66.8** | **63.3** | **57.8** | **60.6** |
| BPF | 69.1 | 68.2 | 68.7 | 60.6 | 63.1 | 62.5 | 68.7 | 56.3 | 62.5 | 71.5 | 53.1 | 66.9 | 62.2 | 48.3 | 55.2 |
| w/ POC | **70.1** | **69.4** | **69.8** | **61.9** | **65.3** | **64.5** | **69.7** | **58.4** | **64.1** | **72.2** | **56.6** | **68.3** | **63.5** | **50.6** | **57.1** |

## C.7 GENERALIZATION ABILITY ANALYSIS

Since our POC only modifies the optimization objective of the classification model, it can be directly integrated into other incremental learning tasks that take classification as a subtask. Therefore, we supplement experiments when POC is integrated into 2 incremental object detection methods, 2 incremental semantic segmentation methods and 1 few-shot incremental instance segmentation method to show the generalization ability of POC. We only integrate POC into the classification branch of these methods, and do not apply the learnable transformations to the "background" category in these tasks since it is a collection of multiple classes.

For the incremental object detection task, we use ABR (Liu et al., 2023) and BPF (Mo et al., 2025) as baseline methods. The mean average precision (mAP) at a 0.5 IoU threshold on the PASCAL-VOC 2007 dataset is reported for comparison. In each incremental setting (A-B), A represents the number of classes in the initial stage, while B indicates the number of classes introduced at each subsequent stage. The results in Table 17 demonstrate improved mAP values across all settings after incorporating POC, highlighting its effectiveness for incremental object detection.

For the incremental semantic segmentation task, we adopt DKD (Baek et al., 2022) and STAR (Chen et al., 2024) as baseline methods. Consistent with the referenced papers, the mean Intersection over Union (mIoU) on the PASCAL-VOC 2012 dataset is reported. In each incremental setting (A-B), A denotes the number of classes in the first stage, and B represents the number of newly introduced classes in each subsequent stage. Table 18 shows improved mIoU values across all configurations after applying POC, underscoring its effectiveness in incremental semantic segmentation.

Table 18: Performance analysis on PASCAL-VOC 2012 to show that POC is generalizable to incremental semantic segmentation task and the mIoU is reported.

| Method | Class Number Settings | | | | | | | | | | | | | | |
|---|---|---|---|---|---|---|---|---|---|---|---|---|---|---|---|
| | 19-1 | | | 15-5 | | | 15-1 | | | 10-1 | | | 5-3 | | |
| | 1-19 | 20 | 1-20 | 1-15 | 16-20 | 1-20 | 1-15 | 16-20 | 1-20 | 1-10 | 11-20 | 1-20 | 1-5 | 6-20 | 1-20 |
| DKD | 78.0 | 57.7 | 77.0 | 79.1 | 60.6 | 74.7 | 78.8 | 52.4 | 72.5 | 74.0 | 56.7 | 65.8 | 69.8 | 60.2 | 62.9 |
| w/ POC | **79.3** | **58.6** | **78.2** | **80.2** | **61.6** | **75.7** | **80.0** | **54.3** | **73.8** | **75.2** | **58.6** | **67.3** | **71.0** | **62.4** | **64.7** |
| STAR | 77.8 | 56.4 | 76.8 | 79.7 | 61.8 | 75.4 | 79.5 | 55.6 | 73.8 | 74.3 | 57.9 | 66.5 | 71.9 | 62.9 | 65.5 |
| w/ POC | **78.7** | **57.2** | **77.6** | **81.7** | **62.7** | **77.1** | **80.4** | **57.6** | **75.1** | **75.6** | **59.2** | **67.8** | **73.4** | **64.8** | **67.4** |

Table 19: Performance analysis on COCO to show that POC is generalizable to few-shot incremental instance segmentation task. The AP and AP50 are reported.

| Shots | Method | Overall | | Base | | Novel | |
|---|---|---|---|---|---|---|---|
| | | AP | AP50 | AP | AP50 | AP | AP50 |
| 1 | iMTFA | 20.1 | 30.6 | 25.9 | 39.3 | 2.8 | 4.7 |
| | w/ POC | **22.2** | **33.6** | **27.8** | **42.4** | **5.2** | **7.3** |
| 5 | iMTFA | 18.2 | 27.1 | 22.5 | 33.2 | 5.1 | 8.6 |
| | w/ POC | **20.9** | **30.0** | **25.3** | **36.4** | **7.7** | **10.6** |
| 10 | iMTFA | 17.8 | 26.4 | 21.8 | 32.0 | 5.8 | 9.8 |
| | w/ POC | **20.1** | **29.1** | **24.1** | **34.7** | **7.9** | **12.4** |

Table 20: Sensitivity analysis of hyper-parameters $\lambda_1$ and $\lambda_2$ in POC on CIFAR-100 under task setting (1). The last accuracy/average incremental accuracy are reported.

| $\lambda_1$ | $\lambda_2$ | | | |
|---|---|---|---|---|
| | 0.01 | 0.1 | 1 | 10 |
| 1 | 56.5/65.1 | 56.9/64.9 | 55.7/64.4 | 54.2/63.1 |
| 10 | 56.1/64.3 | **57.4/65.6** | 56.8/65.1 | 54.1/62.7 |

For the few-shot incremental instance segmentation task, we use iMTFA (Ganea et al., 2021) as the baseline method. Following the referenced work, we report both AP and AP50 on the COCO dataset, considering 60 base classes and 20 novel classes. The results include performances with K=1, 5, and 10 shots per novel class. As shown in Table 19, POC leads to improved AP and AP50 values across all scenarios, further validating its effectiveness for few-shot incremental instance segmentation.

## C.8 HYPERPARAMETER SENSITIVITY ANALYSIS

Based on the experimental results, the hyperparameters $\lambda_1$ and $\lambda_2$ should have a significant impact on the final performance of POC since both transform loss and DCL are crucial components. Therefore, in this section, we conduct experiments to show the sensitivity of the final performance to the hyperparameters. With LUCIR as the baseline method as CIFAR-100 as the evaluation dataset, we choose $\lambda_1$ and $\lambda_2$ from {1,10} and {0.01, 0.1, 1, 10} respectively. The performance of LUCIR with POC is listed in Table 20, showing its sensitivity to hyperparameters.

## C.9 TRAINING COST ANALYSIS

During training, since the model is also trained to classify the augmented samples, the training cost will increase. Here, we design experiments to see whether the training cost will increase a lot. We mainly use the training time to represent training costs. With the same training settings as in Section 4.2.1, we report the GPU days of different methods on CIFAR-100 and ImageNet-100 using one GeForce RTX 3090 GPU. According to the results in Table 21 and Table 22, although the training time increases after adopting our POC, the difference is minor. Furthermore, when the dataset is larger, the proportion of increased time is smaller. This takes advantages of the parallel computing

Table 21: Analysis of training time on CIFAR-100. The GPU days are reported.

| Method | B = 50 | | | B = 20 | | |
|--------|--------|--------|-------|--------|--------|-------|
| | C = 10 | C = 5 | C = 1 | C = 10 | C = 5 | C = 1 |
| LUCIR | 0.23 | 0.26 | 0.31 | 0.27 | 0.32 | 0.39 |
| w/ POC | 0.28 | 0.29 | 0.34 | 0.29 | 0.36 | 0.42 |
| CwD | 0.24 | 0.27 | 0.33 | 0.27 | 0.33 | 0.40 |
| w/ POC | 0.29 | 0.29 | 0.34 | 0.36 | 0.36 | 0.43 |
| PODNet | 0.25 | 0.27 | 0.34 | 0.29 | 0.35 | 0.42 |
| w/ POC | 0.28 | 0.30 | 0.37 | 0.32 | 0.38 | 0.45 |
| MEMO | 0.26 | 0.31 | 0.40 | 0.30 | 0.35 | 0.49 |
| w/ POC | 0.30 | 0.34 | 0.43 | 0.33 | 0.38 | 0.52 |
| LODE | 0.23 | 0.26 | 0.32 | 0.27 | 0.32 | 0.40 |
| w/ POC | 0.28 | 0.29 | 0.34 | 0.29 | 0.36 | 0.42 |
| MRFA | 0.24 | 0.27 | 0.33 | 0.28 | 0.33 | 0.41 |
| w/ POC | 0.29 | 0.30 | 0.36 | 0.32 | 0.36 | 0.45 |

*Class Number Settings*

Table 22: Analysis of training time on ImageNet-100. The GPU days are reported.

| Method | B = 50 | | | B = 20 | | |
|--------|--------|--------|-------|--------|--------|-------|
| | C = 10 | C = 5 | C = 1 | C = 10 | C = 5 | C = 1 |
| LUCIR | 5.14 | 5.36 | 6.52 | 5.83 | 6.48 | 7.70 |
| w/ POC | 5.28 | 5.64 | 6.83 | 5.96 | 6.72 | 8.13 |
| CwD | 5.23 | 5.47 | 6.62 | 5.90 | 6.56 | 7.75 |
| w/ POC | 5.38 | 5.74 | 6.95 | 6.03 | 6.78 | 8.19 |
| PODNet | 6.05 | 6.37 | 7.14 | 6.83 | 7.08 | 8.45 |
| w/ POC | 6.32 | 6.74 | 7.48 | 7.28 | 7.49 | 8.92 |
| MEMO | 6.18 | 6.74 | 7.53 | 6.95 | 7.42 | 9.43 |
| w/ POC | 6.43 | 6.97 | 7.98 | 7.24 | 7.83 | 10.23 |
| LODE | 5.17 | 5.43 | 6.58 | 5.86 | 6.53 | 7.72 |
| w/ POC | 5.34 | 5.69 | 6.90 | 5.98 | 6.75 | 8.15 |
| MRFA | 5.74 | 5.98 | 6.93 | 6.24 | 6.78 | 8.16 |
| w/ POC | 5.98 | 6.14 | 7.21 | 6.74 | 7.12 | 8.46 |

*Class Number Settings*

ability of GPU. Although the batch size increases because of the augmented samples, the parallel computing ensures that the computing time will not increase propositionally to the batch size.

# D    VISUALIZATION RESULTS

This section presents visual representations of the feature distributions for both seen and future classes, aiming to qualitatively demonstrate the efficacy of POC. Using LUCIR with and without POC, we train the model on the initial 5/10 classes of CIFAR-100. Before learning from the following 5 future classes, we visualize the features pertaining to the seen and subsequent 5 classes using T-SNE (van der Maaten & Hinton, 2008). Figure 8 vividly illustrates that with POC integration, future classes are effectively safeguarded from overlapping with seen classes within the model's feature space, demonstrating consistent outcomes across both task settings.

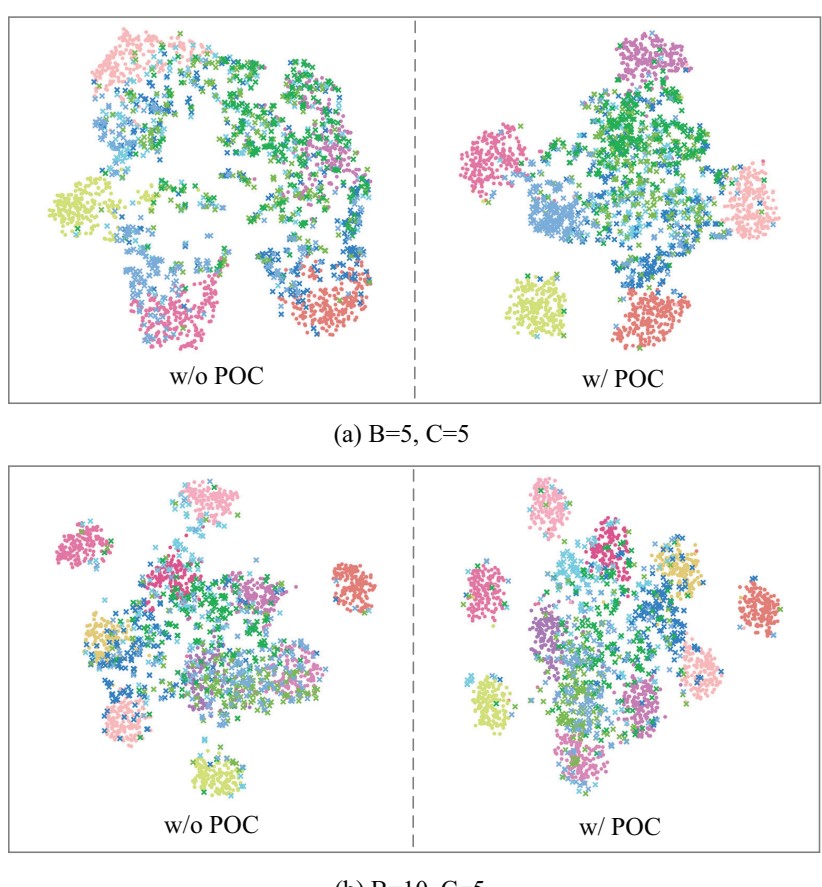

(a) B=5, C=5

(b) B=10, C=5

Figure 8: Illustration of feature distribution for LUCIR with/without POC through T-SNE (van der Maaten & Hinton, 2008). "B" and "C" represent the class number of the first task and the following one task in CIFAR-100. The round points represent the samples from the seen classes and the cross points represent that from the new classes. It is shown that under both task settings, POC can help protect the future classes from overlapping with the seen classes in model's feature space, avoiding catastrophic forgetting.

