# OpenReview forum: "POC: Preventing the Over-Collapse of Classes for Class-Incremental Learning"
_ICLR.cc/2025/Conference — Submitted to ICLR 2025_

### Official Review · Reviewer_UhWh · 2024-10-19

**Soundness:** 3
**Presentation:** 2
**Contribution:** 2
**Rating:** 5
**Confidence:** 4

**Summary:**

This paper aims to improve the generalization on seen classes in CIL by preventing the over-collapse (POC) of seen classes. To this end, the authors generate samples in adjacent regions by some learnable transformations and making the classification model predict them with a modified loss, which is much similar to IL2A where it predicts auxiliary classes generated by mixup. The authors use theories in OOD detection to prove such transformations generate samples in adjacent regions. Furthermore, the author claims the generated samples in adjacent regions are prone to be far away from the seen class, thus introduces the deterministic contrastive loss (DCL) to make them closer to the seen class. Finally, the authors perform performance comparisons with SOTAs and similar works, verify the effectiveness of DCL and POC with ablation study on them and plotting ICD and ICG metrics during the incremental training.

**Strengths:**

- The reported performance of the method out-performs similar works like IL2A.
- The proposed method is compatible with various methods in CIL.

**Weaknesses:**

- The reason why over-collapse leads to forgetting is not clear enough. It seems to assume the samples in the future class are in the adjacent area of the previous samples.
- The DCL is somewhat not well-motivated, there is no empirical or theoretical evidence provided about the _far away_ projection. Instead, the authors state that the distance of the transformed sample is upper-bounded in Proposition 3.2.

**Questions:**

See weaknesses

---

> ### Author Response · Authors · 2024-11-18
>
> Dear Reviewer UhWh,
>
> We are grateful for your valuable time and opinions. We would like to answer your questions according to specific contents in our paper to help you have a more comprehensive understanding of our work.
>
> >Why over-collapse leads to forgetting is not clear.
>
> **The over-collapse indirectly causes forgetting by first leading to overlapping between seen and future classes in the feature space, which will then result in forgetting.** As detailed in line 14, line 46 and Figure 1, over-collapse is a phenomenon that the classification model maps originally separated seen class and its adjacent regions in the input space to be mixed in the feature space. This phenomenon has been observed and analyzed in previous works, such as [1] and [2]. For future classes in these adjacent regions in the input space, the areas covered by them and the seen class after being mapped into feature space will then overlap, as shown both empirically and experimentally in line 46, Figure 3 and Figure 8. As stated in line 53, according to Masana et.al (2022), the overlapping will then result in forgetting since the model will classify the overlapped areas belonging to future classes to minimize classification loss. Therefore, over-collapse will indirectly cause catastrophic forgetting.
>
> **Our approach primarily addresses future classes residing in the adjacent regions of seen classes, as they are more prone to overlapping.** While seen and future classes distant in the input space could also overlap in the feature space, since most neural networks are equivalent to continuous functions and according to [2], it is much more possible for the future classes that are in the adjacent regions of seen class in the input space to overlap with the seen class in the feature space. Therefore, we mainly focus on the future classes residing in the adjacent regions of seen classes in the input space. We will generalize our POC to deal with other possible situations in our future work.
>
> >DCL is not well-motivated and contradictory to Proposition 3.2
>
> DCL is designed to **make the features of one seen class and its adjacent regions close** to protect model’s generalization. Differently, Proposition 3.2 aims to ensure that learnable transformations **produce samples within the adjacent regions of seen class in the input space**. Therefore, **they operate in different spaces with distinct purposes so that are irrelevant and not contradictory to each other**.
>
> **As proved in [3], a classification model will maximize the distance between two classes after being mapped into the feature space** to improve its generalization. In this way, as detailed in line 73 and line 274, **$L_{Mod-Cls}$ will make the model map one seen class and its adjacent regions to be distant in the feature space.** This reduces generalization, as **testing samples in these adjacent regions may cross the classification boundaries, resulting in misclassification**. Therefore, **DCL is proposed to make the features of the seen class and adjacent regions close to protect model’s generalization.** We have added the referenced paper to line 77 for support.
>
> [1] On the decision boundary of deep neural networks
>
> [2] Empirical study of the topology and geometry of deep networks
>
> [3] Large margin deep networks for classification

---

> ### Author Response · Authors · 2024-11-29
> **Gentle Reminder to Reviewer UhWh**
>
> Dear Reviewer UhWh,
>
> Thank you once again for taking the time and effort to review our submission. We sincerely appreciate your valuable feedback and have made a concerted effort to address all your concerns, including clarifying the relationship between over-collapse and forgetting, and the motivation of DCL.
>
> To address your main concerns, we would like to draw your attention to the following results and updates:
>
> - Relationship clarification: **As in line 46, Figure 3 and Figure 8, we both empirically and experimentally demonstrate that over-collapse will lead to overlapping between seen and future classes in the feature space.** And the phenomenon that overlapping between seen and future classes in the feature space will result in forgetting is proved by Masana et.al (2022). **In this way, it is shown that over-collapse will indirectly cause forgetting.** According to [2], the future classes in adjacent regions are more prone to overlapping so that we mainly focus on them.
>
> - Motivation of DCL: According to [3], the model without DCL will map one seen class and its adjacent regions to be distant in the feature space so that the model’s generalization will be harmed. As a solution, **DCL is proposed to make the features of the seen class and adjacent regions close to protect model’s generalization.** Differently, Proposition 3.2 aims to ensure that learnable transformations produce samples within the adjacent regions of seen class in the input space. Therefore, there is no relationship between DCL and Proposition 3.2 so that they are not contradictory to each other.
>
> With the rebuttal period extended by a week, we kindly request your attention to our response. Your assessment is vital to the improvement of our work, and we greatly value your insights. **We are also fully prepared to provide any additional clarifications or analysis you may require to facilitate the re-evaluation of our paper.** Your understanding and guidance mean a great deal to us, and we remain hopeful for your feedback during this extended period.
>
> Best Regards,
>
> The Authors of Submission 5728
>
> [2] Empirical study of the topology and geometry of deep networks
>
> [3] Large margin deep networks for classification

---

> ### Author Response · Authors · 2024-12-02
>
> Dear Reviewer UhWh,
>
> Thank you once again for your valuable comments and suggestions. We have carefully responded to each of your concerns according to the specific lines in our paper and revised our paper based on your suggestions.
>
> We understand that this is a particularly busy time. However, **it is less than 10 hours** to the last time that the reviewers can post message to the authors. Therefore, we deeply appreciate it if you could take a moment to review our responses and let us know if they adequately address your concerns.
>
> Moreover, we would greatly appreciate it if you could re-evaluate the overall score and other aspects of our work based on the responses we have submitted.
>
> Best regards,
>
> The Authors of Submission 5728

---

> > ### Comment · Reviewer_UhWh · 2024-12-02
> >
> > After reading other reviewers' comments and your responses and careful thought, I decided to maintain my rating.

---

### Official Review · Reviewer_rQbH · 2024-10-25

**Soundness:** 2
**Presentation:** 2
**Contribution:** 2
**Rating:** 6
**Confidence:** 3

**Summary:**

This paper tackles the over-collapse phenomenon in class incremental learning (CIL) that makes it difficult to distinguish the seen and unseen classes.
To address this issue, the authors suggest distinguishing the seen classes and their transformed versions.
To generate samples close yet adequately distant from the original ones, the authors suggest rotation and a learnable affine transformation and theoretically demonstrate the effectiveness of these transformations.
The authors argue that the proposed method can prevent over-collapse phenomenon, thereby enhancing generalization ability to unseen classes.

**Strengths:**

The most intriguing aspect is generating samples that are close yet distinct from the original ones.
To prevent the rotated samples from being overly similar to the original ones, the authors propose learning a set of affine transformations, ensuring the generated samples are adequately adjacent but maintain a sufficient distance from the originals.
The theoretical analysis further supports the effectiveness of the proposed transformation for generating adjacent samples.

**Weaknesses:**

It seems that there are no significant issues on the paper.
One minor concern may be the generalization ability of the proposed method.
Can the proposed method be applied to other tasks over the image classification task?
The reviewer thinks it is somewhat difficult to directly generalize the proposed method to other tasks, which may limit its value.

**Questions:**

Please refer to the weakness part.

---

> ### Author Response · Authors · 2024-11-18
>
> Dear Reviewer rQbH,
>
> We sincerely thank you for your valuable comments, which have helped us improve our work.
> In the following, we will solve your concern with more experimental results.
>
> >Whether the method can be applied to other tasks.
>
> Since our POC only modifies the optimization objective of the classification model, **it can be seamlessly integrated into other incremental learning tasks where classification serves as a subtask**. To demonstrate it, we supplement experiments when our POC is integrated into 2 incremental object detection methods, 2 incremental semantic segmentation methods and 1 few-shot incremental instance segmentation method to show the generalization ability of POC. We only apply our POC to the classification branch of these methods, and do not apply the learnable transformations to the “background” category in these tasks since it is a collection of multiple classes.
>
> For the incremental object detection task, we evaluated POC using ABR [1] and BPF [2] as baseline methods on the PASCAL-VOC 2007 dataset. Performance is measured using mean average precision (mAP) at a 0.5 IoU threshold. In each incremental setting A-B(C-D), A represents the number of classes in the initial stage, B indicates the number of classes introduced at each subsequent stage, C-D means the mAP from C-th class to D-th class. **As shown in the table below, incorporating POC consistently improves mAP across all settings, demonstrating its effectiveness in incremental object detection.**
>
> |Method|10-5(1-10)|10-5(11-20)|10-5(1-20)|5-5(1-5)|5-5(6-20)|5-5(1-20)|10-2(1-10)|10-2(11-20)|10-2(1-20)|15-1(1-15)|15-1(16-20)|15-1(1-20)|10-1(1-10)|10-1(11-20)|11-5(1-20)|
> |:-:|:-:|:-:|:-:|:-:|:-:|:-:|:-:|:-:|:-:|:-:|:-:|:-:|:-:|:-:|:-:|
> |ABR|68.7|67.1|67.9|64.7|56.4|58.4|67.0|58.1|62.6|68.7|56.7|65.7|62.0|55.7|58.9|
> |w/ POC|**69.8**|**68.6**|**69.2**|**66.2**|**58.6**|**60.5**|**68.1**|**59.4**|**63.8**|**69.6**|**58.4**|**66.8**|**66.3**|**57.8**|**60.6**|
> |BPF|69.1|68.2|68.7|60.6|63.1|62.5|68.7|56.3|62.5|71.5|53.1|66.9|62.2|48.3|55.2|
> |w/ POC|**70.1**|**69.4**|**69.8**|**61.9**|**65.3**|**64.5**|**69.7**|**58.4**|**64.1**|**72.2**|**56.6**|**68.3**|**63.5**|**50.6**|**57.1**|
>
> For the incremental semantic segmentation task, we adopt DKD [3] and STAR [4] as baseline methods, with performance evaluated using mean Intersection over Union (mIoU) on the PASCAL-VOC 2012 dataset. Similar to the detection task, A-B(C-D) specifies the incremental setting. **The results presented in the table below show consistent improvements in mIoU across all configurations after adopting POC, highlighting its effectiveness in this context.**
>
> |Method|19-1(1-19)|19-1(20)|19-1(1-20)|15-5(1-15)|15-5(16-20)|15-5(1-20)|15-1(1-15)|15-1(16-20)|15-1(1-20)|10-1(1-10)|10-1(11-20)|10-1(1-20)|5-3(1-5)|5-3(6-20)|5-3(1-20)|
> |:-:|:-:|:-:|:-:|:-:|:-:|:-:|:-:|:-:|:-:|:-:|:-:|:-:|:-:|:-:|:-:|
> |DKD|78.0|57.7|77.0|79.1|60.6|74.7|78.8|52.4|72.5|74.0|56.7|65.8|69.8|60.2|62.9|
> |w/ POC|**79.3**|**58.6**|**78.2**|**80.2**|**61.6**|**75.7**|**80.0**|**54.3**|**73.8**|**75.2**|**58.6**|**67.3**|**71.0**|**62.4**|**64.7**|
> |STAR|77.8|56.4|76.8|79.7|61.8|75.4|79.5|55.6|73.8|74.3|57.9|66.5|71.9|62.9|66.5|
> |w/ POC|**78.7**|**57.2**|**77.6**|**81.7**|**62.7**|**77.1**|**80.4**|**57.6**|**75.1**|**75.6**|**59.2**|**67.8**|**73.4**|**64.8**|**67.4**|
>
> For few-shot incremental instance segmentation, we use iMTFA [5] as the baseline method. Following the referenced work, we report both AP and AP50 on the COCO dataset, considering 60 base classes and 20 novel classes. The results include performances with K=1, 5, and 10 shots per novel class. **As shown in the table below, the incorporation of POC leads to improved AP and AP50 values across all scenarios, further validating its effectiveness for few-shot incremental instance segmentation.**
>
>
> |Shots|Method | Overall(AP)|Overall(AP50)|Base(AP)|Base(AP50)|Novel(AP)|Novel(AP50)|
> |:-:|:-:|:-:|:-:|:-:|:-:|:-:|:-:|
> |1|iMTFA|20.1|30.6|25.9|39.3|2.8|4.7|
> |1|w/ POC|**22.2**|**33.6**|**27.8**|**42.4**|**5.2**|**7.3**|
> |5|iMTFA|18.2|27.1|22.5|33.2|5.1|8.6|
> |5|w/ POC|**20.9**|**30.0**|**25.3**|**36.4**|**7.7**|**10.6**|
> |10|iMTFA|17.8|26.4|21.8|32.0|5.8|9.8|
> |10|w/ POC|**20.1**|**29.1**|**24.1**|**34.7**|**7.9**|**12.4**|
>
>
> In conclusion, our POC is generalizable to other incremental learning tasks where classification serves as a subtask. We have supplemented the experiments into Section C.7 of our new version of paper.
>
> [1] Augmented box replay: Overcoming foreground shift for incremental object detection
>
> [2] Bridge past and future: Overcoming information asymmetry in incremental object detection
>
> [3] Decomposed knowledge distillation for class-incremental semantic segmentation
>
> [4] Saving 100x storage: prototype replay for reconstructing training sample distribution in class-incremental semantic segmentation
>
> [5] Incremental few-shot instance segmentation

---

> ### Author Response · Authors · 2024-11-29
> **Gentle Reminder to Reviewer rQbH**
>
> Dear Reviewer rQbH,
>
> Thank you once again for taking the time to review our submission. We deeply appreciate your thoughtful feedback and have worked diligently to address your problem by supplementing more experiments and analysis on other incremental learning tasks.
>
> To address your primary concern, we would like to highlight the following results and updates:
>
> - Generalize POC to other tasks: We integrate our POC into 2 incremental object detection methods, 2 incremental semantic segmentation methods and 1 few-shot incremental instance segmentation method to show that POC is generalizable to other incremental learning tasks where classification serves as a subtask. The results supplemented in Table 17, 18 and 19 in Section C.7 show that **incorporating POC consistently improves performance across all tasks and settings, demonstrating its effectiveness in other incremental learning tasks.**
>
> With the rebuttal period extended by a week, we kindly request your attention to our response. Your evaluation is crucial to improving our work, and we greatly value your insights. **We are fully committed to addressing any questions or providing additional analysis that might assist in re-evaluating our paper.** Your understanding and guidance are deeply appreciated, and we remain hopeful for your feedback during this extended period.
>
> Best Regards,
>
> The Authors of Submission 5728

---

> > ### Comment · Reviewer_rQbH · 2024-11-29
> >
> > Dear authors,
> >
> > Thank you for your detailed response.
> > Your response has clearly addressed my concerns about generalization to other tasks.
> > After carefully reading all other reviewers' comments and your responses, I decided to maintain the original positive rating.
> >
> > Best Regards,
> > The reviewer rQbH

---

> > > ### Author Response · Authors · 2024-11-29
> > > **Gratitude to Reviewer rQbH**
> > >
> > > Dear Reviewer rQbH,
> > >
> > > Thank you very much for supporting our POC! We are glad to know that our response has addressed your concerns.
> > >
> > > We appreciate the discussion, your feedback, and your suggestions. Many thanks for your time and effort.
> > >
> > > Best regards,
> > >
> > > The Authors of Submission 5728

---

### Official Review · Reviewer_dVqr · 2024-11-04

**Soundness:** 2
**Presentation:** 2
**Contribution:** 2
**Rating:** 5
**Confidence:** 5

**Summary:**

the authors propose a two-step framework called Prevent the Over-Collapse (POC) for class incremental learning. During training, POC applies transformations to training samples of seen classes, maintaining their distinction in the feature space. It also introduces an expanded classifier to separate seen classes from adjacent regions. In the testing phase, the expanded classifier is masked, allowing classification of seen classes without extra computational costs. POC incorporates a deterministic contrastive loss to keep adjacent regions close to their original classes, enhancing generalization. Experimental results on CIFAR-100 and ImageNet show that POC improves the last and average incremental accuracy of several state-of-the-art CIL methods by 3.5% and 3.0%, respectively.

**Strengths:**

-	The proposed POC framework effectively prevents the overlap between seen and future classes in the feature space as shown in fig.3. This innovative approach might enhance the model's ability to generalize across tasks.
-	The experimental results show that POC can robustly enhance the performance of various CIL approaches across several approaches.
-	The article provides sufficient evidence for some of its claims in the appendix.

**Weaknesses:**

-	An important assumption of the POC is that it addresses the issue of over-collapse, which can lead to catastrophic forgetting. However, there is insufficient literature to prove that over-collapse is the cause of catastrophic forgetting. The citations provided in the article, such as Masana et al., 2022 on line 184, do not offer relevant explanations, and the article also does not sufficiently analyze the over-collapse phenomenon as claimed in its contributions. This results in the article appearing to lack a reasonable motivation for its claims.
-	The POC requires inference on multiple augmented images, which may lead to a significant increase in training costs. However, the article does not discuss this issue.
-	I’m not certain that the primary reason POC is effective is due to its backbone having seen multiple augmented images. I believe it is necessary to conduct an experiment where all images augmented by learnable augmentations are used as positive samples corresponding to their categories for direct training. I think this approach could also yield some performance improvement, and it might not perform worse than the results shown by POC.

**Questions:**

All my concerns are mentioned in the weakness. And I think the experiments in the third line of weakness should be conducted.

---

> ### Author Response · Authors · 2024-11-18
>
> Dear Reviewer dVqr,
>
> We sincerely appreciate your valuable feedback and the opportunity to clarify our work. Below, we provide detailed responses to your concerns, supported by additional explanations and supplemented experimental results.
>
> > Insufficient analysis of over-collapse and its relationship with catastrophic forgetting.
>
> **In our paper, we primarily demonstrate that over-collapse will lead to overlapping between seen and future classes in the feature space. By citing previous work (Marana et.al 2022) to prove that overlapping can lead to catastrophic forgetting, we can indirectly show that over-collapse is one of the causes of forgetting. And as in line 46, Figure 3 and Figure 8, we both empirically and experimentally show that over-collapse will lead to overlapping, supporting our motivation to prevent over-collapse to avoid catastrophic forgetting.**
>
> Regarding the link between overlapping and catastrophic forgetting, as outlined in lines 12 and 53, when only limited samples from seen classes are available during the learning of future classes, the classification model tends to misidentify overlapping samples from seen classes as belonging to future classes. This behavior minimizes classification loss but results in knowledge forgetting. **This relationship has been previously demonstrated in works such as Masana et al. (2022) so that we cite it for support, rather than to claim the relationship between over-collapse and forgetting**. We apologize for the ambiguity in line 184 of the original text and have revised it in our new version of paper.
>
> The over-collapse phenomenon is analogous to the well-known generalization behavior of classification models in other tasks, and has been analyzed in previous studies. For instance, **references [1] and [2] reveal that models will extend their decision boundaries beyond the distribution of training data, mapping one class and its adjacent regions in the input space to be mixed in the feature space to improve generalization.** While this behavior is beneficial in other classification tasks, we find it detrimental in class-incremental learning. To emphasize its negative impact, we rename the phenomenon. We have added the two references to line 49 for support.
>
> We provide both empirical and experimental evidence to establish the relevance between over-collapse and overlapping. Empirically, as shown in line 50 and Figure 1, over-collapse causes the classification model to map one seen class and future classes in its adjacent regions to overlap within the feature space. Experimentally, baselines adopting POC demonstrate reduced ICD (as shown in Figure 3), which quantitatively measures the extent of overlapping between seen and future classes. Additionally, the visualization in Figure 8 illustrates the separation of seen and future classes after adopting POC, qualitatively supporting the same conclusion. **These results collectively show the relevance between over-collapse and overlapping so that over-collapse will indirectly cause forgetting, validating our motivation to prevent over-collapse.**
>
> [1] On the decision boundary of deep neural networks
>
> [2] Empirical study of the topology and geometry of deep networks

---

> ### Author Response · Authors · 2024-11-18
>
> > Discuss the issue of increased training costs
>
> To address concerns regarding training costs, we primarily use training time as a representative metric. With the same experimental settings as in Section 4.2.1, we report the training GPU days of various methods on CIFAR-100 and ImageNet, utilizing a single GeForce RTX 3090 GPU. The tables below show the results on CIFAR-100 and ImageNet-100 respectively. **As shown in the table, while adopting our POC slightly increases the training time, the difference remains minimal. Furthermore, when the dataset is larger, the proportion of increased time is smaller.**. This is due to the parallel computing capability of GPU. Although the batch size increases due to the classification of augmented images, **parallel computation ensures that the overall training time does not scale proportionally with the batch size.** We have supplemented the experiments in Section C.9 in new version of paper.
> |Method | B=50, C=10 | B=50, C=5 | B=50, C=1 | B=20, C=10 | B=20, C=5 | B=20, C=1|
> |:--------: |:--------:|:--------:|:--------:| :--------:|:--------:|:--------:|
> |LUCIR|0.23|0.26|0.31|0.27|0.32|0.39|
> |w/ POC|0.28|0.29|0.34|0.29|0.36|0.42|
> |CwD|0.24|0.27|0.33|0.27|0.33|0.40|
> |w/ POC|0.29|0.29|0.34|0.36|0.36|0.43|
> |PODNet|0.25|0.27|0.34|0.29|0.35|0.42|
> |w/ POC|0.28|0.30|0.37|0.32|0.38|0.45|
> |MEMO|0.26|0.31|0.40|0.30|0.35|0.49|
> |w/ POC|0.30|0.34|0.43|0.33|0.38|0.52|
> |LODE|0.23|0.26|0.32|0.27|0.32|0.40|
> |w/ POC|0.28|0.29|0.34|0.29|0.36|0.42|
> |MRFA|0.24|0.27|0.33|0.28|0.33|0.41|
> |w/ POC|0.29|0.30|0.36|0.32|0.36|0.45|
>
> |Method | B=50, C=10 | B=50, C=5 | B=50, C=1 | B=20, C=10 | B=20, C=5 | B=20, C=1|
> |:--------: |:--------:|:--------:|:--------:| :--------:|:--------:|:--------:|
> |LUCIR|5.14|5.36|6.52|5.83|6.48|7.70|
> |w/ POC|5.23|5.64|6.83|5.96|6.72|8.13|
> |CwD|5.23|5.47|6.62|5.90|6.56|7.75|
> |w/ POC|5.38|5.74|6.95|6.03|6.78|8.19|
> |PODNet|6.05|6.37|7.14|6.83|7.08|8.45|
> |w/ POC|6.32|6.74|7.48|7.28|7.49|8.92|
> |MEMO|6.18|6.74|7.53|6.95|7.42|9.43|
> |w/ POC|6.43|6.97|7.98|7.24|7.83|10.23|
> |LODE|5.17|5.43|6.58|5.86|6.53|7.72|
> |w/ POC|5.34|5.69|6.90|5.98|6.75|8.15|
> |MRFA|5.74|5.98|6.93|6.24|6.78|8.16|
> |w/ POC|5.98|6.14|7.21|6.74|7.12|8.46|

---

> ### Author Response · Authors · 2024-11-18
>
> >Compare with results when augmented images are used as positive samples.
>
> Following the setting of Section 4.2.1, we evaluate the performance on CIFAR-100 and ImageNet-100 when images produced by transformations are used as positive samples. The tables below are the results on CIFAR-100 and ImageNet-100 respectively. **According to the results, the performance gain is lower than that when the augmented samples are regarded as negative**.
>
> We analyze the Inter-Class Distance (ICD) and Intra-Class Generalization (ICG), and the results are in Figure 4 in new version of paper. The results show that when augmented samples are considered positive, both ICD and ICG increases. It results from that rotated samples have similar distribution to the original ones as discussed in line 258. Additionally, the transform loss $L_{Trans}$ makes the transformations similar to rotations. Therefore, the transformations will converge to rotations to minimize both $L_{Mod-Cls}$ and $L_{Trans}$, reducing $L_{Total}$. This effectively mirrors training with rotation augmentations so that the decision boundary of one class will extend more broadly. **While this enhances model's generalization, it exacerbates the over-collapse phenomenon, which in turn worsens the overlap between seen and future classes. As a result, the overall performance gain is limited. Instead, POC is designed to both prevent over-collapse and protect the generalization so that it obtains better performance.** The experiments have been supplemented in Section C.2 in new version of paper.
>
> |Method|B=50, C=10|B=50, C=5|B=50, C=1|B=20, C=10|B=20, C=5|B=20, C=1|
> |:-:|:-:|:-:|:-:|:-:|:-:|:-:|
> |LUCIR|64.1$\pm$ 0.9|61.2$\pm$ 0.7|55.9$\pm$ 0.3|59.4$\pm$ 0.5|57.6$\pm$ 0.3|48.5$\pm$ 0.2|
> |Positive|64.9$\pm$ 1.0|61.8$\pm$ 0.8|56.7$\pm$ 0.3|60.5$\pm$ 0.7|58.2$\pm$ 0.2|49.6$\pm$ 0.5|
> |Negative|**66.8$\pm$ 0.7**|**63.5$\pm$ 0.6**|**59.6$\pm$ 0.4**|**63.8$\pm$ 0.3**|**59.2$\pm$ 0.3**|**53.1$\pm$ 0.2**|
> |CwD|67.2$\pm$ 0.2|62.8$\pm$ 0.1|59.7$\pm$ 0.2|64.3$\pm$ 0.4|61.2$\pm$ 0.5|53.6$\pm$ 0.3|
> |Positive|68.0$\pm$ 0.4|63.8$\pm$ 0.1|60.6$\pm$ 0.4|65.1$\pm$ 0.5|62.1$\pm$ 0.3|54.7$\pm$ 0.4|
> |Negative|**69.6$\pm$ 0.4**|**65.4$\pm$ 0.3**|**62.3$\pm$ 0.5**|**68.3$\pm$ 0.2**|**66.1$\pm$ 0.4**|**59.1$\pm$ 0.2**|
> |PODNet|64.6$\pm$ 0.7|63.2$\pm$ 1.1|59.8$\pm$ 0.5|54.9$\pm$ 0.4|53.2$\pm$ 0.4|50.5$\pm$ 0.2|
> |Positive|65.2$\pm$ 0.6|63.7$\pm$ 1.1|60.5$\pm$ 0.7|56.0$\pm$ 0.6|54.0$\pm$ 0.6|51.2$\pm$ 0.4|
> |Negative|**68.2$\pm$ 0.8**|**67.2$\pm$ 1.0**|**63.1$\pm$ 0.7**|**60.6$\pm$ 0.7**|**58.3$\pm$ 0.4**|**53.5$\pm$ 0.5**|
> |MEMO|70.2$\pm$ 0.5|69.0$\pm$ 0.7|61.4$\pm$ 0.3|69.5$\pm$ 0.5|67.3$\pm$ 0.8|63.2$\pm$ 0.4|
> |Positive|70.8$\pm$ 0.7|69.6$\pm$ 0.6|62.0$\pm$ 0.5|70.0$\pm$ 0.5|67.9$\pm$ 0.5|63.7$\pm$ 0.4|
> |Negative|**71.8$\pm$ 0.6**|**70.4$\pm$ 0.4**|**63.5$\pm$ 0.5**|**70.9$\pm$ 0.6**|**69.3$\pm$ 0.4**|**64.8$\pm$ 0.6**|
> |LODE|68.7$\pm$ 0.6|64.6$\pm$ 0.8|58.5$\pm$ 0.4|66.2$\pm$ 0.5|64.4$\pm$ 0.3|59.2$\pm$ 0.5|
> |Positive|69.2$\pm$ 0.7|65.1$\pm$ 0.5|58.9$\pm$ 0.5|67.0$\pm$ 0.6|64.8$\pm$ 0.4|59.7$\pm$ 0.6|
> |Negative|**70.0$\pm$ 0.5**|**66.1$\pm$ 0.7**|**60.5$\pm$ 0.7**|**68.4$\pm$ 0.3**|**65.8$\pm$ 0.6**|**62.4$\pm$ 0.4**|
> |MRFA|68.0$\pm$ 0.4|66.4$\pm$ 0.6|60.3$\pm$ 0.8|67.8$\pm$ 0.8|65.7$\pm$ 0.6|61.3$\pm$ 0.7|
> |Positive|68.5$\pm$ 0.5|66.8$\pm$ 0.5|60.8$\pm$ 0.7|68.4$\pm$ 0.7|66.2$\pm$ 0.6|62.0$\pm$ 0.5|
> |Negative|**69.2$\pm$ 0.4**|**68.1$\pm$ 0.5**|**62.7$\pm$ 0.5**|**69.6$\pm$ 0.6**|**67.5$\pm$ 0.4**|**63.6$\pm$ 0.8**|
>
> |Method|B=50, C=10|B=50, C=5|B=50, C=1|B=20, C=10|B=20, C=5|B=20, C=1|
> |:-:|:-:|:-:|:-:|:-:|:-:|:-:|
> |LUCIR|61.4/71.5|55.1/67.2|41.1/56.8|48.0/61.5|42.6/55.7|34.3/48.9|
> |Positive|62.1/72.3|55.8/67.6|42.1/57.7|48.9/62.3|43.4/56.6|34.8/49.4|
> |Negative|**64.0/73.7**|**57.6/68.3**|**47.7/61.8**|**51.5/65.2**|**46.4/59.3**|**36.9/51.5**|
> |CwD|60.4/71.6|55.8/68.2|40.3/56.3|48.2/62.9|44.6/58.5|34.3/51.1|
> |Positive|60.9/72.2|56.3/68.6|40.8/56.5|49.0/63.7|45.3/59.1|34.9/51.7|
> |Negative|**62.3/73.2**|**57.4/69.4**|**44.7/59.8**|**51.2/64.4**|**47.1/60.6**|**38.9/53.1**|
> |PODNet|62.3/73.4|57.4/71.6|42.9/59.7|45.8/63.0|41.7/59.8|32.4/50.0|
> |Positive|62.8/73.8|57.9/72.3|43.7/60.4|46.4/63.9|42.2/60.4|32.9/50.7|
> |Negative|**63.8/75.0**|**62.3/72.8**|**48.6/63.7**|**49.1/64.8**|**48.2/62.1**|**36.6/55.1**|
> |MEMO|66.2/76.8|64.5/76.4|52.7/64.0|53.6/67.1|48.4/60.8|40.3/53.2|
> |Positive|66.7/77.3|65.0/76.8|53.4/64.6|54.3/67.7|49.1/61.7|40.8/53.9|
> |Negative|**67.4/77.9**|**66.5/77.8**|**55.9/66.5**|**55.4/68.2**|**50.7/62.5**|**42.4/54.7**|
> |LODE|64.5/73.6|59.4/71.0|45.8/60.4|50.6/63.5|45.3/59.5|37.2/52.1|
> |Positive|64.8/74.0|59.7/71.5|46.3/60.8|51.0/64.0|45.9/59.9|37.8/52.6|
> |Negative|**66.1/75.1**|**61.7/73.1**|**50.4/63.8**|**53.4/65.7**|**48.3/62.3**|**40.5/53.4**|
> |MRFA|65.1/74.8|61.4/73.2|47.3/61.6|51.8/64.9|46.1/60.0|38.5/52.6|
> |Positive|65.7/75.2|61.7/73.6|47.9/62.3|52.3/65.3|46.6/60.5|38.9/53.2|
> |Negative|**66.4/76.0**|**63.3/74.9**|**50.6/64.4**|**54.1/66.6**|**48.7/62.1**|**41.1/54.3**|

---

> ### Author Response · Authors · 2024-11-29
> **Gentle Reminder to Reviewer dVqr**
>
> Dear Reviewer dVqr,
>
> Thank you once again for your time and effort in reviewing our submission. We sincerely appreciate your valuable feedback and have worked diligently to address all your concerns, including clarifying our motivation, developing training time analysis, comparing with positive sampling and revising the manuscript accordingly.
>
> To address your main concerns, we would like to draw your attention to the following results and updates:
>
> - Motivation clarification: **We have modified line 49 and line 184 to clarify our motivation to prevent the over-collapse, which is that it will lead to overlapping between seen and future classes.** This can be shown both empirically and experimentally by line 46, Figure 3 and Figure 8. By citing the work of Masana et.al (2022) to prove that overlapping can result in catastrophic forgetting, **we indirectly show that over-collapse is one of the causes of forgetting, supporting our motivation.**
>
> - Training time analysis: We supplement the training time analysis of POC with the results in Table 21 and 22 in Section C.9. Thanks to the parallel computing ability of GPU, **the increased training time of backbones after adopting POC is minor and less than 10% in most cases.**
>
> - Compare with positive sampling: We supplement the performance on CIFAR-100 and ImageNet-100 when images produced by learnable transformations are used as positive samples. The results in Table 11, 12 and Figure 4 in Section C.2 show that **although the generalization of models is enhanced when using positive sampling, the over-collapse is exacerbated so that the performance gain is minor.**
>
> As the rebuttal period has been extended by a week, we humbly and kindly request your attention to our response. Your assessment is critical to the improvement of our work, and we deeply value your insights. **Moreover, we remain fully committed to providing any clarifications or additional analysis you might find necessary to help re-evaluate our paper.** Your understanding and guidance would mean a lot to us, and we remain hopeful for your response during this extended period.
>
> Best Regards,
>
> The Authors of Submission 5728

---

> ### Author Response · Authors · 2024-12-02
>
> Dear Reviewer dVqr,
>
> Thank you once again for your valuable comments and suggestions. We have carefully responded to each of your concerns, provided additional experimental results to further clarify our points, and revised our paper based on your suggestions.
>
> We understand that this is a particularly busy time. However, **it is less than 10 hours** to the last time that the reviewers can post message to the authors. Therefore, we deeply appreciate it if you could take a moment to review our responses and let us know if they adequately address your concerns.
>
> Moreover, we would greatly appreciate it if you could re-evaluate the overall score and other aspects of our work based on the responses we have submitted.
>
> Best regards,
>
> The Authors of Submission 5728

---

### Author Response · Authors · 2024-11-18
**Gratitude to All the Reviewers**

We would like to express our sincere gratitude to all the reviewers for their careful and thoughtful review of our manuscript, as well as for the valuable feedback they have provided. Their comments have contributed to the improvement of our paper and have deepened our understanding of the subject. We have uploaded our responses to each reviewer's comments, along with a revised version of the paper, which we hope adequately addresses the points raised.

---

### Author Response · Authors · 2024-11-25

Dear Reviewers,

Thank you once again for your insightful comments and suggestions—they have been immensely helpful in improving our work. We have carefully addressed each of your points and included additional experimental results to clarify and strengthen our arguments.

We understand this may be a particularly busy time, and we deeply appreciate your taking the time to review our responses. Please let us know if our revisions adequately address your concerns. Should you have any further feedback, we will do our utmost to address it promptly.

Best regards,

The Authors of Submission 5728

---

### Meta-Review · Area_Chair_GACc · 2024-12-18

**Metareview:**

This submission received two negative sores and a positive score after rebuttal. After carefully reading the paper, the review comments, the AC can not recommend the acceptance of this submission, as the average score is under the threshold bar and the concerns about the motivation and justification of the proposed approach remain. The AC recognizes the contributions confirmed by the reviewers, and encourages authors to update the paper according to the discussion and submit it to the upcoming conference.

**Additional Comments On Reviewer Discussion:**

After discussion, reviewer#rQbH thought that the response had clearly addressed her/his concerns about generalization to other tasks and kept a positive rating. Reviewer# dVqr stated that the motivation is still not clear enough and maintained a negative rating. Reviewer#UhWh read the responses, but she/he decided to maintain a negative rating.

---

### Decision · Program_Chairs · 2025-01-22

Reject